# An objective identification technique for potential vorticity structures associated with African Easterly Waves

Christoph Fischer[1,2,3], Andreas H. Fink[2], Elmar Schömer[3], Marc Rautenhaus[1], and Michael Riemer[4]

[1]Regional Computing Centre, Visual Data Analysis Group, Universität Hamburg, Hamburg, Germany
[2]Institute of Meteorology and Climate Research, Karlsruhe Institute of Technology, Karlsruhe, Germany
[3]Institute of Computer Science, Johannes-Gutenberg University, Mainz, Germany
[4]Institute for Atmospheric Physics, Johannes-Gutenberg University, Mainz, Germany

**Correspondence:** Christoph Fischer (christoph.fischer@uni-hamburg.de)

**Abstract.**

Tropical Africa and the North Atlantic Ocean are significantly influenced by African Easterly Waves (AEWs), which play a fundamental role in tropical rainfall and cyclogenesis in that region. The dynamics of AEWs can be described in a potential vorticity (PV) framework. The important impact of latent heat release by cloud processes is captured in this framework by the diabatic generation of PV anomalies. This paper introduces an innovative approach for the identification and tracking of PV structures within AEWs. By employing AEW tracking and computing the wave phase of each point within the AEW domain using a Hilbert transform, we are able to effectively identify and collect 3-D PV structures associated with specific AEWs. To facilitate a climatological analysis, here performed over the months June to October from 2002 to 2022, these structures are subsequently characterized by low-dimensional descriptors, including their location, intensity, and orientation.

Our climatological analysis reveals the seasonal evolution and the structural attributes of PV anomalies within AEWs over the study domain. PV feature locations closely align with the African Easterly Jet's latitudinal shift during the summer season. Analysis of the mean pressure level of the 3-D PV structures shows a remarkable shift during their life cycle, indicating deep moist convection characteristics over land, and more shallow convection characteristics over the ocean. On average, PV features identified within AEW troughs tilt downshear over land and equatorward over the ocean. The trough-centered analysis reveals distinct differences between satellite-estimated and model-predicted rainfall. Agreement between the results of a more traditional composite analysis and our new feature analysis provides confidence in our feature approach as a novel diagnostic tool. The feature framework provides a low-dimensional representation of AEWs' PV structure, which facilitates future statistical analyses of the relation of this structure to, e.g., tropical cyclogenesis or predictability of tropical rainfall.

## 1 Introduction

African Easterly Waves (AEWs) are synoptic-scale disturbances that play a crucial role in the weather and climate of tropical West Africa and the tropical Atlantic region. AEWs are quasi-periodic perturbations, typically originating over the broader Lake Chad region in central North Africa, or being triggered by the high topography over the Ethiopian Highlands (Mekonnen et al., 2006; Hamilton et al., 2020). These waves propagate westward across West Africa, the North Atlantic Ocean and as far as the

eastern Pacific. They have drawn considerable attention due to their substantial impact on Atlantic tropical cyclone (TC) genesis (e.g., Russell et al., 2017; Núñez Ocasio, 2021; Rajasree et al., 2023), rainfall variability over the West African monsoon region, their relation to the West African offshore rainfall maximum (e.g., Hamilton et al., 2017), and their role in extreme precipitation events over tropical West Africa (e.g., Fink and Reiner, 2003; Crétat et al., 2015; Engel et al., 2017). Understanding the formation, propagation, and interaction of AEWs with the ambient (thermo-)dynamical state of the troposphere is essential for improving the prediction of North Atlantic TCs and rainfall patterns in West Africa.

The potential vorticity (PV) framework is a fundamental fluid-dynamical conceptual model widely utilized in extratropical meteorology, including the understanding of barotropic and baroclinic instabilities, Rossby wave propagation and amplification (e.g., Hoskins et al., 1985), Rossby wave breaking (McIntyre and Palmer, 1983; Thorncroft et al., 1993), and the importance of latent heat release and other diabatic processes on Rossby wave dynamics (e.g., Teubler and Riemer, 2021). Furthermore, under a balance assumption, the wind, temperature and density fields can be derived solely from the PV field. Strong latent heat release associated with convection or intense rainfall (Weijenborg et al., 2017; Müller et al., 2020) leads to horizontal and vertical dipoles of PV, and thus rich small-scale structures. Müller et al. (2020) highlighted the correlation of strong PV anomalies and intense precipitation and suggested that PV anomalies may serve as a proxy for evaluating intense rainfall. In the absence of non-conservative processes, PV is materially conserved, which in combination with large PV gradients in the mid-latitudes makes it relatively straightforward to identify and track PV features associated with Rossby waves and other large scale flow features (e.g., Teubler and Riemer, 2021; Fischer et al., 2022; Hauser et al., 2023). In the tropics, however, much smaller PV gradients and more prominent contributions of convective-scale latent heat release imply a more complex and smaller-scale nature of the PV distribution, which is in particular true for AEWs. A feature-based PV perspective of AEWs thus faces challenges that demand a more detailed investigation.

There exists a substantial body of research on AEWs, especially encompassing their dynamical interaction with the environment and their relationship with TCs. The dry dynamics of AEWs can be understood in terms of downstream propagation along the African Easterly Jet (AEJ) from an upstream wave source (Thorncroft et al., 2008), with (small) amplification by baroclinic and barotropic growth (Hall et al., 2006). As in the midlatitudes, these processes can be described from the PV perspective. More important for AEW amplification is latent heat release associated with embedded convection (Berry and Thorncroft, 2005; Thorncroft et al., 2008). From a PV perspective, this amplification is seen as the diabatic generation of PV anomalies. Besides amplitude, the diabatically generated PV anomalies signify modification of AEW structure. Tomassini et al. (2017) investigated in detail the contributions of different parameterization schemes to diabatically modified PV in an operational numerical model. Russell et al. (2017, 2020) give further insight into the structure and sources of PV in AEWs, including the role of moist convection and its coupling with the background wave environment. Essentially, the diabatically generated PV anomalies encapsulate the impact of moist processes on AEW intensity and structure that outlasts a period of active convection.

In terms of AEW predictability, there is strong indication that uncertainty in latent heat release can be linked to larger forecast errors in AEW characteristics (Elless and Torn, 2018, 2019). Recently, Núñez Ocasio et al. (2020) suggested that the type of convective organization and the location of the convection relative to the AEW trough may play a discriminating role in terms

of AEW-related TC genesis in the North Atlantic Ocean. Dunkerton et al. (2009) provide a conceptual framework that links AEWs and TC genesis. In this framework, the nonlinear critical layer of AEWs provides a region of recirculating air masses, in which moisture and PV may accumulate with time in relative isolation from the drier and thus more hostile environment. Despite these potential applications in understanding AEW growth, predictability and their role in TC genesis, the application of a PV-centric view, especially in three dimensions, in studying AEWs has been relatively limited.

In this study, we address this gap by providing a comprehensive tool for the PV-centric analysis of AEWs. Specifically, we address the following objectives:

- We propose a novel identification and tracking technique for 3-D PV features associated with AEWs. It facilitates the quantification of feature properties in case studies and climatological analyses. By describing features by a low-dimensional vector, statistical analyses including feature climatologies (e.g., Limbach et al., 2012), ensemble forecast analysis (e.g., Rautenhaus et al., 2015a), and statistical postprocessing (e.g., Rasp and Lerch, 2018) can be performed.

- We perform a climatological analysis of these identified PV features to explore the properties of these features over their life-cycle. Furthermore, it provides confidence in the previously introduced method by comparing the identified features to climatological PV composites.

Multiple approaches exist for objectively identifying AEWs. Early approaches by Burpee (1972), Reed et al. (1988), and Diedhiou et al. (1999) focus on the identification of mean tracks rather than individual wave anomalies. In order to be able to compute climatologies and to perform statistical analyses of the data, an objective identification is absolutely vital. Objective identification techniques for atmospheric features have proven beneficial in both atmospheric research for statistical analyses and verification tasks, and in operational meteorology (e.g. Hengstebeck et al., 2011). Fink and Reiner (2003) summarize the strength and weaknesses of automatic and manual approaches. Thorncroft and Hodges (2001) were the first ones to exploit the usefulness of the vorticity measure to objectively track individual AEWs in the tropics. Other approaches include the identification using Hovmöller diagrams (Bain et al., 2014). Berry et al. (2007) and Belanger et al. (2016) most recently used the advection of curvature vorticity as primary measure of AEW activity and to identify wave tracks. This measure proved to be very robust after applying downscaling and smoothing operators.

To enhance our analysis of areas with high PV associated with AEWs, we incorporate phase filtering through the Hilbert transform. This technique is essential because the geographic location of an AEW trough often does not match the regions with the highest PV anomalies, due to the complex interactions between wave movements and diabatic PV influenced by convection. According to Shapiro (1978) and Tomassini et al. (2017), convection within AEWs typically begins ahead, or west, of the trough over Africa, where atmospheric conditions are most conducive to convection (Reed et al., 1977; Fink and Reiner, 2003). Conversely, over the ocean, convection generally occurs closer to, or slightly east of, the trough (Riehl, 1954). Since convection can potentially take place across the entire trough area, accurately assigning the current phase to every point in the domain becomes crucial for filtering PV signals in AEWs.

To reach the aforementioned goals of this work, we employ a technique similar to the one by Belanger et al. (2016) to get a robust position of AEWs on 700 hPa, this is outlined in Sect. 2.2 and 2.3. Then, in Sect. 2.4, we compute the wave

phase at every point in the domain by performing a Hilbert transform, which is required to narrow down the regions around AEWs where high PV associated with these AEWs is expected (Fink and Reiner, 2003). Then finally, in Sect. 2.5, we will identify and extract the PV structures. The identified 3-D features are assigned geometric descriptions as low-dimensional representation, including a best-fitting ellipsoid. Section 3 shows results of a climatological analysis of this features, shedding light on the distinct characteristics of the waves. Additionally, the approach is being verified through comparison with PV composites. A trough-centered analysis highlights differences between satellite-estimated and model-predicted rainfall. The paper is concluded in Sect. 4, where we will discuss the implications of our findings, their relevance to previous research, and potential avenues for further investigation.

## 2   Strategy

### 2.1   Data

For the identification of AEWs and the corresponding PV features, as well as for the analyses in this study, we utilize data from the global ERA-5 reanalysis (Hersbach et al., 2020). Our analysis focuses on the period from June to October to align with the West African Monsoon season, as detailed by Fink et al. (2017). The selected data is provided on a regular grid with a grid-point spacing of 0.5° in both latitude and longitude, with a temporal resolution of 6 hours. This time scale is sufficient since AEW characteristics do not vary on hourly timescales. To identify the wave trough of AEWs, we use the zonal (u) and meridional (v) wind components at 700 hPa, where the AEJ has its maximum speed (Fink et al., 2017). The study domain covers the region of AEW activity over tropical West Africa and parts of the North Atlantic Ocean, specifically from 75°W to 45°E and 0°N to 40°N. The PV analysis is conducted on 16 pressure levels between 200 and 900 hPa, with intervals of 50 hPa. The PV data on pressure levels originates from the ERA-5 archive. We exclude the pressure levels beneath 900 hPa where surface-induced effects come into play. For rainfall analyses, the satellite-gauge based GPM IMERG V06B data set (Huffman et al., 2015) and the twice-daily ERA-5 short-range forecasts will be used to quantify rainfall in relation to AEWs and PV occurrence.

### 2.2   Identification of AEW wave troughs

To robustly identify PV features within AEWs, we build on the method by Belanger et al. (2016) to firstly identify AEW troughs on 700 hPa. Their method is an improvement over previous work (e.g., Thorncroft and Hodges, 2001) by applying curvature vorticity (CV) anomalies instead of absolute values of relative vorticity and by ensuring that waves are westward propagating, more closely aligning with the characteristics of AEWs. We have improved the tracking of the waves to achieve more robust and consistent tracks. Additionally, we introduce a data structure to facilitate the analysis of the tracks, including the identification of split and merge events, computation of average wave speeds, and other parameters.

A climatology of CV is computed for each month and each time of day to take into account both seasonal and diurnal effects. Then, the CV itself is bandpass-filtered to only retain frequencies that are linked to AEW disturbances. We use a filter to keep

frequencies between 2–8 days, following Russell et al. (2020). To identify wave troughs at a given point in time, we compute the anomalies of the CV (CVA) by subtracting this bandpass-filtered CV from the climatology. In the idealized case of westward-propagating waves, positive CVAs are linked to cyclonic rotation present in the wave trough, while negative CVAs are present in the ridge regions. Fig. 1a shows an example of CV anomalies. The bandpass-filtered wind leads to distinct alternating positive and negative poles of CVA. Following Belanger et al. (2016), the data is smoothed twice using a 9-point local smoother, which retains only the synoptic-scale easterly wave structure. Then, the zeros of the CVA advection are determined, which resemble the lines where the sign of the advection changes. These are the troughs and ridges in the CVA field. Given the grid's discrete nature, a cell wise computation would almost never find zero-values. Hence, we employ the Marching Squares algorithm to interpolate line segments between grid cells that approximate these zero-lines. These segments are then merged to form continuous lines. In Fig. 1a, the grey lines indicate zeros of CVA advection, which collocate with troughs and ridges.

For filtering purposes, following Belanger et al. (2016), two masks are applied to the identified lines:

- the zonal wind must be $u < +2.5\mathrm{ms}^{-1}$,

- and CVA must be over the 66-th percentile of the entire reanalysis.

Furthermore, we consult second derivative of the CVA to extract only troughs in the data set, masking out the ridges. The red lines in Fig. 1a are the result of applying the other masks and filters to the data set. This results in the identified wave troughs.

## 2.3 Tracking of AEW wave troughs

Research on tracking AEWs has produced various methodologies, where the most established ones are based on analyzing the CV field. Hollis et al. (2024), for example, utilize an approach based on the well-known TRACK algorithm, originally proposed by Hodges (1995). This method primarily focuses on linking point features across successive time frames based on a predefined, physically reasonable propagation speed. Lawton et al. (2022) adopt a different approach by tracking AEWs through meridional averages of CV and velocity. Bain et al. (2014) and Brammer and Thorncroft (2015) employ Hovmöller plots for their tracking, analyzing the longitudinal movement of waves. While each of these methodologies has proven effective in their respective applications and has gained popularity in the field, they predominantly focus on point features. In contrast, our work, along with that of Belanger et al. (2016), leverages additional information provided by the identified wave trough features.

To form tracks from the identified individual wave troughs, we employ an overlap approach. Overlap tracking has proven to be a robust tracking technique in meteorological applications, such as tracking of Mesoscale Convective Systems (e.g., Núñez Ocasio and Moon, 2024; Feng et al., 2023; Prein et al., 2023) and general purpose feature extraction (e.g., Ullrich et al., 2021). However, since our identified wave troughs are represented as line strings, they don't directly lend themselves to traditional overlap tracking methods.

To address this, we create area features by predicting the future positions of each trough for upcoming time steps, $t + \Delta t$ and $t + 2\Delta t$, with $\Delta t = 6\,\mathrm{h}$. This prediction uses an anticipated propagation speed to define a polygonal area that represents where the trough is expected to be. The presence of overlap between these predicted polygonal areas and the actual locations

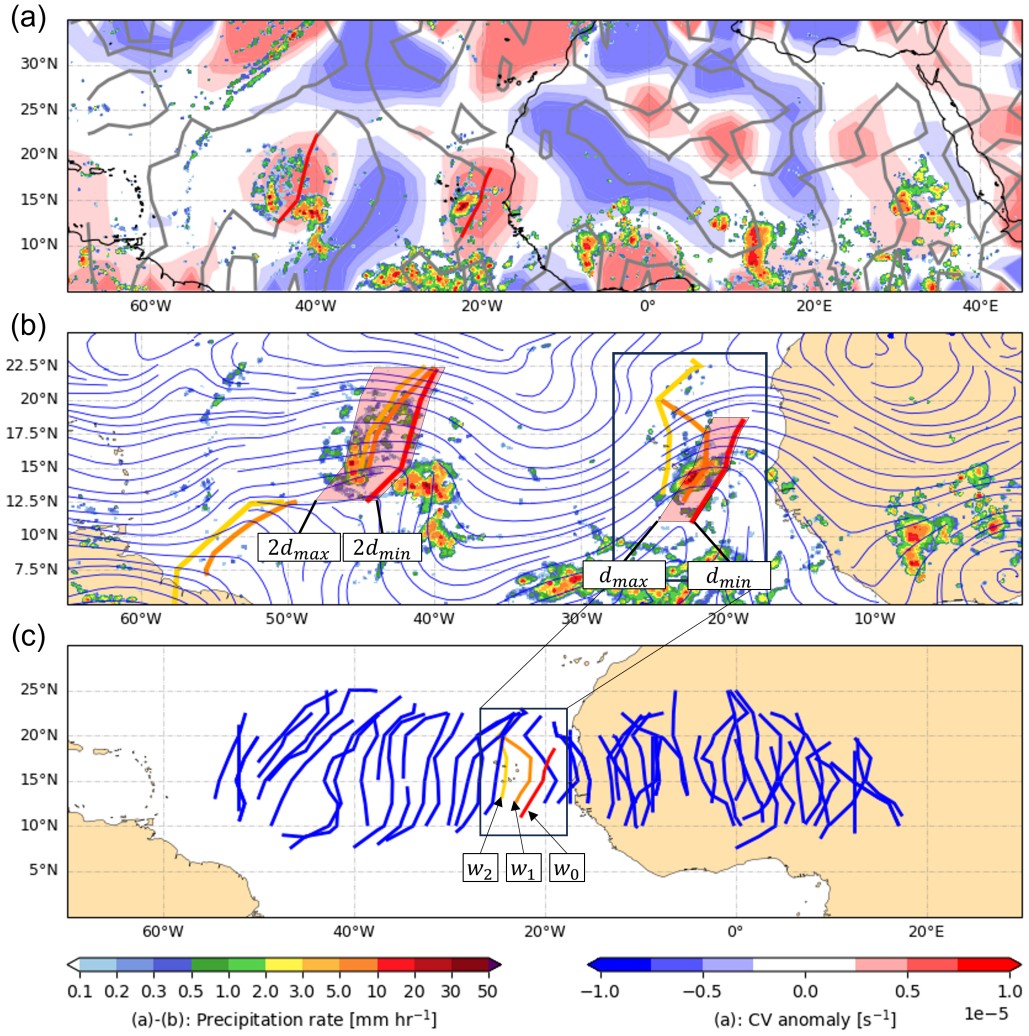

**Figure 1.** Illustration of the identification and tracking process for AEWs using a polygonal search approach. (a) shows the CV anomalies and the precipitation rate according to the GPM IMERG data set at 13 September 2022, 00 UTC. Grey lines indicate the zeros of CVA advection, and the red lines show the extracted wave troughs based on the filters introduced in Sect. 2.2. In (b), two identified wave troughs are shown at 00 UTC (red, as in (a)), 06 UTC (orange), and 12 UTC (yellow). Red boxes sketch the polygonal search areas computed for both wave troughs, initiated at 00 UTC. The left wave trough is compared with the +12 h wave trough, while the right wave trough is compared with the +6 h wave trough. The 700 hPa streamlines of the bandpass-filtered wind at 00 UTC are depicted as blue lines. In panel (c), an entire AEW track is depicted, demonstrating the continuity of the tracking process. The three wave troughs from (b) are highlighted.

of wave troughs at future time steps facilitates the tracking. The range of expected propagation speeds, which we set from $u_{max} = -15\,\mathrm{ms}^{-1}$ to $u_{min} = +2\,\mathrm{ms}^{-1}$, defines the size and shape of these polygons, as illustrated in Fig. 1b. While AEWs typically propagate westward with a speed between $5\,\mathrm{ms}^{-1}$ and $10\,\mathrm{ms}^{-1}$ (Gu et al., 2004), we use a broader range to account for sampling artifacts due to the data resolution. Nonetheless, based on the expected wavelength of AEWs (e.g., Fink et al., 2017), this range still ensures we do not mix up different waves into the same track. Given a time granularity $\Delta t$, the bounds of expected traveled distance $d_{min}$ and $d_{max}$ can be computed by multiplying $u_{min}$ and $u_{max}$ with the elapsed time.

With this approach, wave tracks can persist even when a trough is detected at $t$ and $t + 2\Delta t$, but not at $t + \Delta t$. This happens, when one of the above-mentioned identification masks is not fulfilled for this time step, but for the ones before and after. Consequently, maintaining longer continuous wave tracks, rather than multiple short tracks, enhances the quality of the tracking results.

Using the identified connections, a connection graph $G = (W, E)$ is formed, akin to the approach outlined by Limbach et al. (2012). This structure enables the application of graph theory concepts and algorithms to our identified features, as also employed by Whitehall et al. (2015) in their analysis of MCS. $W$ represents the set of all wave troughs and serves as the nodes in the graph, while $E$ denotes the edges connecting elements of $W$. A pair of wave troughs $w_1, w_2 \in W$ forms an edge $(w_1, w_2) \in E$ if they represent the same entity, thus, fulfilling the overlap strategy outlined above. This graph provides a concise representation of the union of all wave troughs and their evolution over time. For instance, if there exist two wave troughs $w_1, w_2$ following wave trough $w$, such that $(w, w_1) \in E$ and $(w, w_2) \in E$, the wave trough splits in two, indicating a split event. The connection graph is being simplified by removing transitive edges that are primarily introduced through the comparison of wave troughs from non-consecutive time steps. Thus, if all $\{(w_1, w_2), (w_2, w_3), (w_1, w_3)\} \subseteq E$, then $(w_1, w_3)$ is eliminated from $E$.

After the connection graph has been simplified, tracks are extracted from the graph. Each track $T^i$ itself is a graph $T^i = (W^i, E^i)$, based on a subset of wave troughs $W^i \subseteq W$ and connections $E^i \subseteq E$. The tracks $T^i$ are disjoint subsets of the overall connection graph $G$. The extraction of tracks poses challenges: During a wave trough's life cycle, it interacts with the dynamic environment and might split into multiple parts (e.g., due to a weakening at the center part of the trough), and potentially merge again later. Therefore, different sets of tracks can be justified as a solution to this problem. Here, we adhere to the rule to generate tracks with the longest possible lifespan. This makes it easier to investigate the life cycle of these waves. Alternative heuristics, such as extracting connected sub-graphs or initiating new tracks at every split and merge event, were considered. Extracting connected sub-graphs could reduce the number of tracks and inherently capture split and merge events within a single track. However, this approach is sensitive to inaccuracies in feature identification and overlooks the origins of merged features, connecting too many features into one. On the other hand, creating new tracks at each split or merge event significantly increases the number of tracks, making life span analyses unfeasible.

To generate tracks with the longest possible lifespan, the graph $G$ is scanned and each node is assigned a time until dissipation. Then, tracks are formed starting from the nodes with the longest time until dissipation. In case of split events, only the track with the longest time until dissipation is continued, the other sub-track forms a different track. Subsequently, the extracted tracks $T^i$ can be individually analyzed and filtered. For each identified wave trough being part of a track, we compute

its current speed as the average speed over a 2-day window centered around the trough's time step along this track. For this purpose, the position of a wave trough is defined by the center of its bounding box. We discard parts of tracks with an average speed of less than $3 \, \mathrm{ms}^{-1}$ at any given point in time. This removes stationary features based on orographic effects. Furthermore following Belanger et al. (2016), we also remove tracks with a lifetime of less than 2 days from the track set.

Figure 1c shows the full life-cycle of one tracked AEW from (b). Each AEW trough represents a node in the graph. The troughs of the depicted track are linked using edges: The $w_0$ manifestation of the wave is connected to $w_1$, thus $(w_0, w_1) \in E$. $w_1$ itself is connected downstream to the next trough $w_2$, and so on. When the track splits, both ways can be continued in this way. We refer to Limbach et al. (2012) for a detailed introduction on this type of data structure.

The output of the tracking algorithm consists of a list of AEW troughs with unique identifiers and their descriptions, and a list of tracks where each track is defined by a list of edges defining that track. The location of the AEW is defined as a line string, thus a list of latitude-longitude points. The computed AEW trough data set, spanning the entire ERA-5 reanalysis period from 1940 to 2022, is available as detailed in the data availability section. Also, a near real-time implementation displaying identified waves from the ECMWF, GFS and ICON deterministic forecasts is available[1].

## 2.4 Phase Computation

Although an AEW trough defines the line of maximum CV, its location does not necessarily coincide with the expected region of maximum PV anomalies, which could occur in the entire trough phase. This discrepancy underscores the need to compute the wave phase for each point across the domain. Looking at the meridional wind component $v$ in the background flow, the trough area can be assigned a phase between $-\pi$ to $0$, and the ridge a phase between $0$ and $\pi$. However, it is not straightforward to extract the phase information from a real-valued field, since all frequencies in the bandpass-filtered range of 2–8 days can contribute to it.

Zimin et al. (2003) faced a similar problem and utilized digital signal processing methods to extract amplitude and phase information of Rossby wave packets. Using the so-called Hilbert transform (see a practical introduction in Purves, 2014), a complex (consisting of a real and imaginary) field can be computed from a real-valued field, which provides additional information such as amplitude and phase. This method involves performing a Fourier transform on the signal and then applying a back-transform only to the positive part of the frequency spectrum. This approach is suitable for wave patterns that encompass a range of wavenumbers, like tropical waves do. Following Zimin et al. (2003), we look at the meridional wind component $v$ along a latitude circle and perform a 2-D Discrete Fourier Transform over both the wavenumber and time domains:

$$V_{f,k} = \sum_{t=0}^{T-1} \sum_{n=0}^{N-1} v(t,n) e^{-2\pi j \left( f \frac{t}{T} + k \frac{n}{N} \right)}, \tag{1}$$

where $t$ denotes the time step in the data set consisting of $T$ time steps, and $n$ the longitude index out of $N$ indices, while $f$ and $k$ denote the frequency and wavenumber coefficients of the transformed signal. Following Reed et al. (1977) and Russell et al. (2020), we choose a wavelength of 2000–6500 km and a wave frequency of 2–8 days to identify perturbations that can

---

[1]www.kit-weather.de/aew_deterministic_maps.php (last accessed at 15 Feb 2024)

be assigned to AEWs. Therefore, the back transformation is only applied for the subset the positive wavenumbers $k_{min} \leq k \leq k_{max}$ and positive frequencies $f_{min} \leq f \leq f_{max}$ corresponding to the chosen wavelength and frequency criteria:

$$v_H(t,n) = \frac{2}{TN} \sum_{f=f_{min}}^{f_{max}} \sum_{k=k_{min}}^{k_{max}} V_{f,k} e^{2\pi j (f\frac{t}{T} + l\frac{n}{N})}. \tag{2}$$

This process can be described as a bandpass-filter that generates complex wave information. The real part of the restored signal represents the bandpass-filtered data. Additionally, the imaginary part allows for the computation of the phase at any point in the domain using the equation:

$$\theta(v_H) = arctan(\frac{Im[v_H]}{Re[v_H]}). \tag{3}$$

Figure 2 provides an illustration of a wind field at 700 hPa, along with the identified wave troughs highlighted in red. To remove phase signals in the trough area $(-\pi \ldots 0)$ that are not near any AEW trough at all (e.g., vorticity anomalies from other tropical and extratropical waves), we restrict the validity of the phase to a radius of 1000 km around the trough line, depicted by the red contour in Fig. 2. Since the transform is performed for wavelength greater than 2000 km, this ensures we consider a substantial area that does not interfere with signals of other waves. The shading represents the phase of the wave at each point in the domain periodically from $-\pi$ to $\pi$. The wave troughs clearly align with a phase of $-\frac{\pi}{2}$ (white areas). The hatched area in Fig. 2 represents the designated search area for PV, called "trough area" from hereon, which is determined by the areas in the trough phase and near a wavetrough (within the red radius).

This highlights that the Hilbert transform produces correct results with regard to the expected wave signal. In areas less conducive to AEW activity, the Hilbert transform's reliability diminishes, leading to increased noise and less accurate results in the phase field. This outcome is anticipated, as different types of atmospheric waves and other atmospheric disturbances overshadow the AEW signal in these locations. In addition, the green iso-contours in Fig. 2 indicate the vertically averaged PV in the lower and mid troposphere, highlighting the regions of high PV. Notably, these high PV areas align well with the phase and proximity to the wave troughs.

## 2.5   Identification of PV features

These previously identified trough areas in Sect. 2.4 delineate the search area for PV. Analyses of the PV field (e.g., Fig. 3), supported by 3-D analyses using Met.3D (Rautenhaus et al., 2015b), have indicated that a threshold of 0.7 PVU retains the strong signal associated with the waves while effectively separating the clusters. Therefore, we identify all grid points within trough areas that have a PV value exceeding 0.7 PVU. Figure 3a displays a raw 3-D PV field in the tropics, visualized using the 0.7 PVU iso-contour. The noisy PV field highlights the need for a procedure to extract PV that is linked to AEWs. In Fig. 3b, the PV areas over this threshold areas have been confined based on the trough areas defined in Sect. 2.4, and displayed as yellow volumetric objects. Additionally, a so-called morphological opening filter got applied to these objects to refine the features. A comprehensive overview of these morphological filters is provided by Najman and Talbot (2013), with a recent implementation for extratropical PV features applied in Fischer et al. (2022). These filters process the 3-D volumetric features

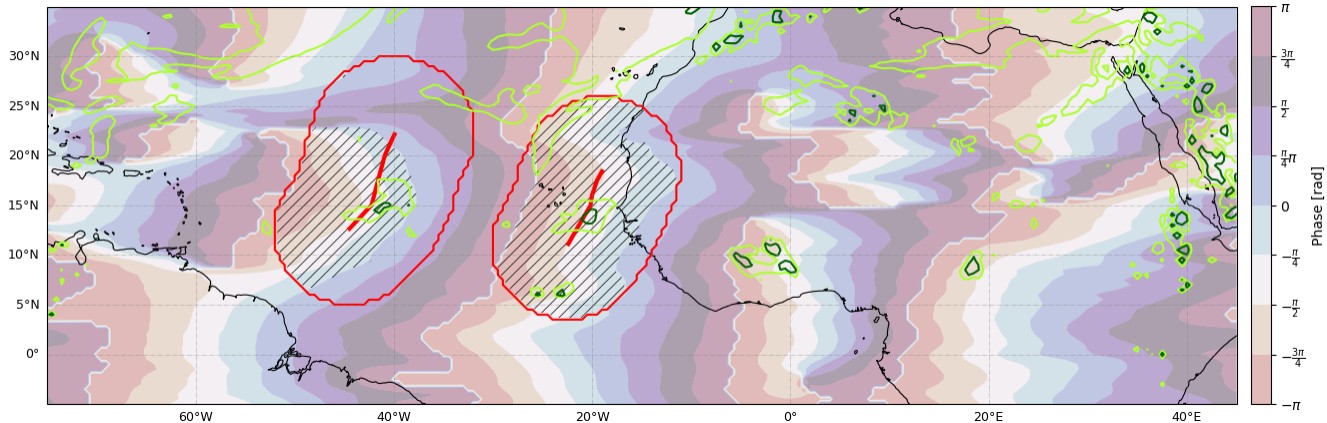

**Figure 2.** Visualization of the computed wave phase using a Hilbert transform (shading), and the identified wave troughs with a 1000 km radius in red at 13 September 2022, 00 UTC. The hatched region represents the intersection of the expected phase and the vicinity area of the wave trough, delineating the search area for high PV. High PV regions are indicated by the green iso-contours (light green 0.4 PVU, dark green 0.8 PVU), which are based on the averaged PV values between 900 and 300 hPa.

by eliminating small, isolated outliers and smoothing out noisy areas, while preserving their general structure. Based on the
proximity to the 2-D wave trough lines, each PV feature can be linked to such a wave trough. Furthermore, the tracks of the 3-D PV anomalies can be inferred from the tracks of the 2-D wave troughs, as introduced in Sect. 2.3. A set of feature descriptions is computed for each PV feature, consisting of bounding box, maximum and average PV value, volume, and as outlined next, a set of image moments and a geometric representation.

## 2.6 Ellipsoid Computation

To facilitate statistical analyses of the identified features, it is desirable to represent them using a set of intuitive and well-suited parameters. Previous studies have revealed that the structure and orientation of PV features in AEWs depend on different factors, for example their location and the (thermo-)dynamic constraints of the environment (Tomassini et al., 2017; Russell et al., 2020; Núñez Ocasio and Rios-Berrios, 2023). Ellipsoids are deformed spheres and can be defined by their center and their three main axes. This simple representation encapsulates information about the shape's position, size, orientation, and elonga-
tion, while being defined by only four vectors. It has been termed as the "most economical representation" for meteorological observations by Smith and Woolf (1976), is also employed in various other scientific fields, including robotics and metrology.

Various approaches have been considered for computing a best-fit ellipsoid. Minimizing the squared error between the surface of the feature and the ellipsoid is computationally expensive and problematic on discrete grids (see an overview in Turner et al., 1999). An ellipsoid with the minimum volume that fully encloses the PV feature can be computed using Khachiyan's al-
gorithm (Khachiyan, 1996), but this approach is not robust against noise. We have determined that the most suitable method for computing ellipsoids is to calculate so-called image moments of the feature. Image moments have been widely used in image

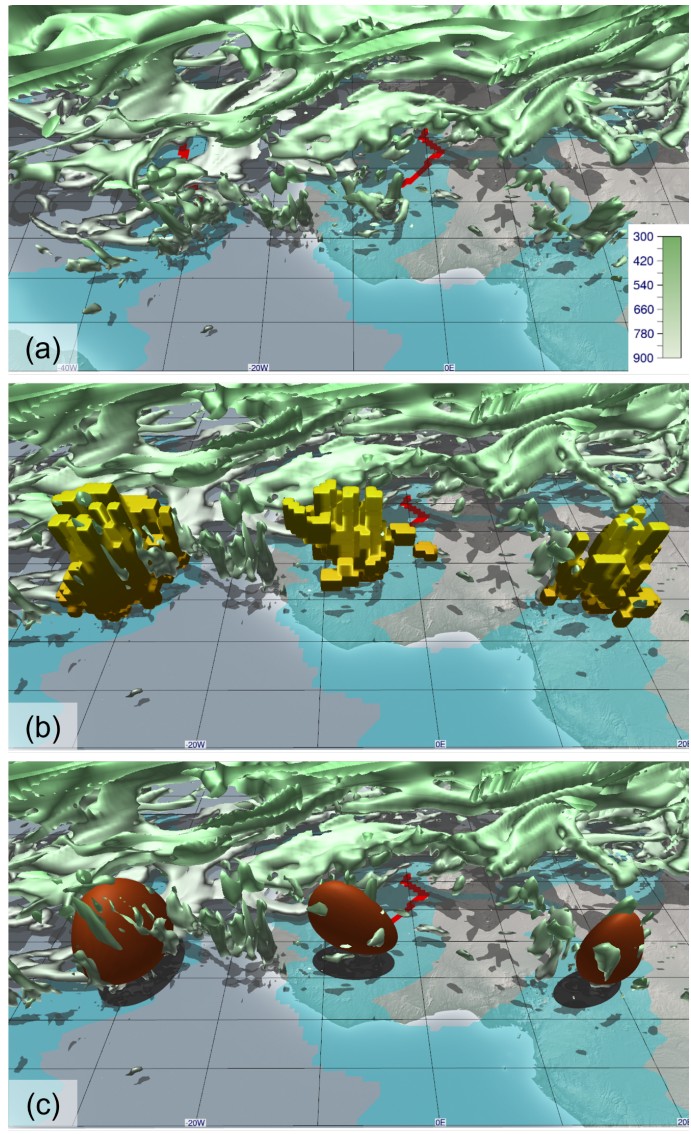

**Figure 3.** A 3-D visualization of the PV across the North Atlantic Ocean and West Africa at 13 September 2022, 00 UTC, created using Met.3D, is depicted. In (a), the 0.7 PVU iso-contour is displayed with shading corresponding to atmospheric pressure, identified wave troughs in red, and the trough phase ranging from $-\pi$ to 0 is highlighted in blue. (b) showcases the identified PV objects in yellow as outlined in Sect. 2.5, while (c) illustrates the ellipsoids representing the identified features from (b).

analysis and pattern recognition (e.g., Teague, 1980; Papakostas et al., 2010) as they compactly describe the spatial character-istics and geometric properties of a feature. We refer to Mukundan and Ramakrishnan (1998) for a detailed introduction of this concept. Importantly, we will exploit the second-order moments to compute the main orientation of a feature.

For a discrete setting, as for our grid, the 3-D moments $M_{ijk}$ of order $i + j + k$ for the function $PV$ are defined as

$$M_{ijk} = \sum_x \sum_y \sum_z x^i y^j z^k PV(x,y,z) V(x,y,z), \tag{4}$$

where $PV$ represents the PV restricted to the identified volumetric feature (e.g., yellow structure in Fig. 3b), $V$ the volume of the grid cell, and $x$, $y$, and $z$ are the grid dimensions (longitude, latitude, pressure levels). Calculating the moments up to the second order provides multiple interesting characteristics of the shape. The 0th moment $M_{000}$ represents the PV-weighted

volume of the object, while the vector

$$(x_c, y_c, z_c) = (\frac{M_{100}}{M_{000}}, \frac{M_{010}}{M_{000}}, \frac{M_{001}}{M_{000}}) \tag{5}$$

represents the centroid of the feature weighted by PV. The second order moments represent the variance and covariance between each pair of dimensions. Therefore, we can construct the covariance matrix $\mathbf{\Sigma}$ of the feature as

$$\mathbf{\Sigma} = \begin{bmatrix} \mu'_{200} & \mu'_{110} & \mu'_{101} \\ \mu'_{110} & \mu'_{020} & \mu'_{011} \\ \mu'_{101} & \mu'_{011} & \mu'_{002} \end{bmatrix}, \tag{6}$$

where $\mu'_{ijk} = \frac{M_{ijk}}{M_{000}} - x_c^i y_c^j z_c^k$ are the second order central moments. As demonstrated in Jackson (2005), the eigenvectors of the covariance matrix $\mathbf{\Sigma}$ correspond to the main axes of the feature, while the length of the main axes $s_i$ can be computed from the eigenvalues $\lambda_i$ as $s_i = 2\sqrt{\lambda_i}$. These principal axes are determined for each PV feature, defining the best-fit ellipsoid by spanning these three axes. An example is illustrated in Fig. 3c, where the ellipsoids are the ones computed for the yellow objects in (b). Furthermore, ellipsoids are invariant under projections: Projecting ellipsoids on any 2-D plane yields ellipses

(see Hartley and Zisserman, 2003), which in turn can be characterized by two main axes. We compute the projected ellipses on the three axis-aligned planes ($xy$, $xz$, and $yz$ plane), and define these ellipses by their main axes.

## 3    Climatology

To assess the data set generated by the identification algorithm and provide an overview of the features, a climatological analysis is conducted. Figure 4 presents a climatology depicting the occurrence of wave troughs and geometrical representations. In

this context, occurrence represents the fraction of time steps within a given month where a grid point resides within the trough area introduced in Sect. 2.4, indicated by the colored shading. The climatological mean zonal wind at 700 hPa, representing the AEJ, exhibits a latitudinal movement and coincides with regions of higher wave trough occurrence. Additionally, for specific longitudes, the histogram in Fig. 4f displays the amount of identified wave troughs centered at each longitude. It indicates whether a geometric representation, as defined in Sect. 2.6, has been assigned to the respective feature. Errors during the

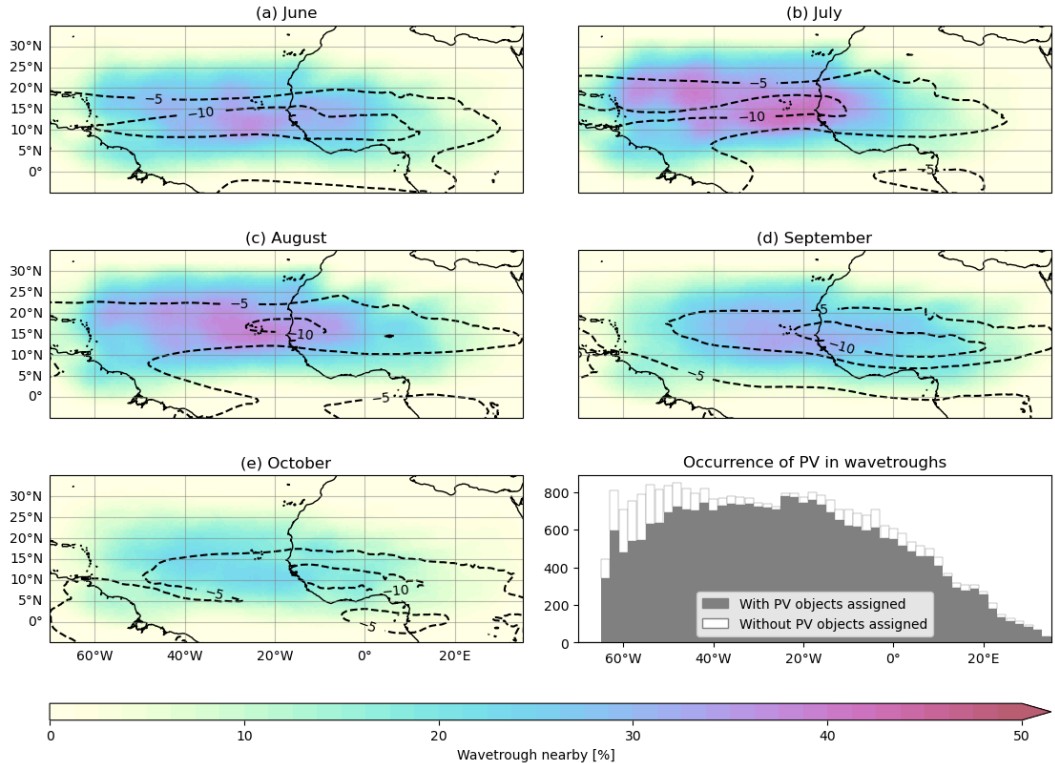

**Figure 4.** Occurrence percentages of trough areas for the months of (a) June to (e) October at a specific point, along with iso-lines depicting mean zonal wind at 700 hPa (-5 and -10 ms$^{-1}$). (f) shows the number of identified wave troughs at various longitudes based on their bounding box center, and indicates the ratio of wave troughs which have been assigned a geometric 3-D PV representation. The ERA-5 reanalysis of 2002 to 2022 has been used.

computation of the ellipsoid (e.g., due to an insufficient amount of data points exceeding the PV threshold) may prevent the generation of a geometric representation for a PV feature. The plot reveals that in regions where well-developed AEWs are expected, the majority of these waves exhibit a distinct PV signal. However, the PV signal weakens in the genesis region of northern East Africa due to the lack of deep moist convection, and the western end of the study domain, where almost half of the waves moving towards northern South America are accompanied with inhibition of precipitation (Giraldo-Cardenas et al.,

305 2022).

Additionally, Fig. 5 illustrates the changes in PV feature occurrence between the months shown in Fig. 4. The mount of PV features increase from June to July, corresponding to the West African Monsoon season's onset. Warmer and moister atmospheric conditions favor PV anomaly formation. Then, going into August, the PV activity increases further over tropical West Africa, aligning with peak convective activity during the monsoon season and the peak of wave activity in general

(Fink et al., 2017). Inversely to panel (a), September shows a decline in PV activity, mainly over the Atlantic Ocean, as the

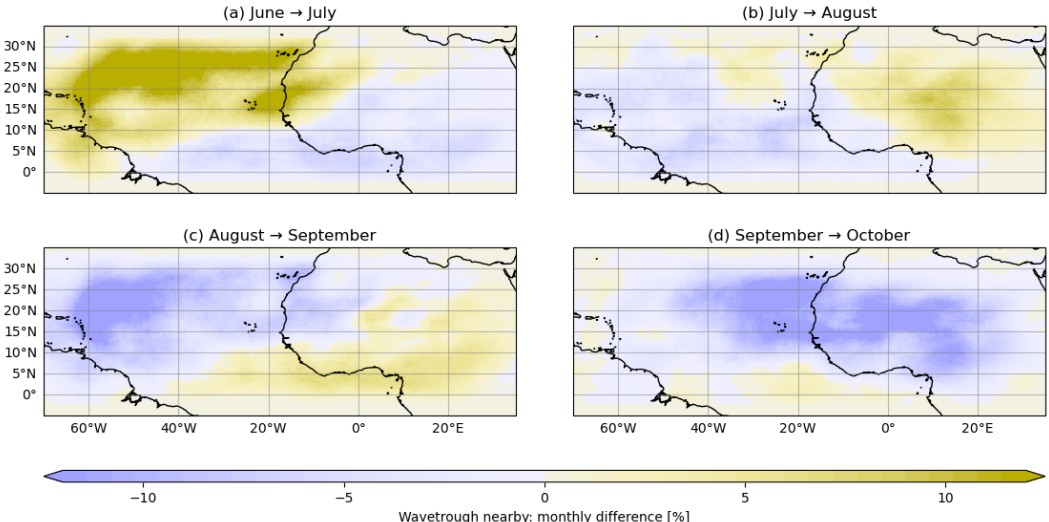

**Figure 5.** Differences in occurrence percentage of consecutive months in the June–October season, based on the monthly data from Fig. 4.

Intertropical Convergence Zone (ITCZ) shifts southward. Towards October, PV activity over West Africa retreats southward as well in tandem with the monsoon's withdrawal.

As 3-D reference, Figure 6 shows a composite of the PV anomalies relative to the identified wave troughs, categorized in *Ocean* and *Land*. The categorization is based on whether the midpoint of the wave trough's bounding box is situated over tropical West Africa (Land) or the Atlantic Ocean (Ocean). For each grid point along each identified wave trough being part of a track, the data is split into bins based on the longitude offset ahead and behind the trough. This leads to a zonal cross section of the climatologically average AEW, looking from the south into it. Additionally, below the 3-D composites, the estimated rainfall (based on GPM-IMERG satellite retrievals), and the ERA-5 modeled rainfall is shown. The latter one is an 18-hour forecast initialized twice a day (06 UTC, 18 UTC) with hourly data available. The lead times from +6 h to +18 h are concatenated to avoid model spin up effects. Then, both the forecast's and satellite retrieval's data is centered to 6-hourly around the given wave trough time to get a estimate of the rainfall rates at a given location by both data sources.

Over land, the PV column exhibits a noticeable downshear tilt (the lower tropospheric shear vector points westward due to the AEJ), whereas over the ocean, the PV column appears upright in relation to this cross-section. Centered around the wave trough, a clear dipole structure in the meridional wind can be observed, with a maximum around 650-700 hPa, falling in line with the maximum intensity of the AEJ (e.g., Burpee, 1972). Moreover, over land high PV values extend higher into the troposphere than over ocean. The composite structure over land is reminiscent of the deep convection that occurs ahead of the trough (Fink and Reiner, 2003). This is corroborated by the ERA-5 model rainfall that peaks slightly ahead of the trough. Over land clear differences occur, with higher estimated rainfall intensities well ahead of the trough line. As discussed in Fink and Reiner (2003), squall lines are initiated ahead of the trough, but move about twice as fast as the trough into the preceding ridge where they tend to dissipate. This is one potential explanation of the rainfall intensities increasing west of the trough in the

composite. Over ocean, the PV tower is shallower and centered on the trough. The shallower PV object over ocean is related to less deep convection over ocean due to lower CAPE values. The largest model rainfall is found at and to the east of the trough axis. This is consistent with an AEW evolution described in Riehl (1954) and Russell et al. (2020). The satellite-based rainfall estimations match the model rainfall in phase and amplitude quite well over ocean.

In Figure 7, a similar composite is presented from an other viewing angle. It shows the latitude-pressure cross-section perspective, viewing from east to west into the PV anomalies. To get a composite of PV structures independent on their current latitude, we use here the computed center of the ellipsoid as reference latitude in the center (red line). Areas to the left indicate locations south of the center of the PV anomaly, and areas to the right denote areas north. This view validates the PV feature locations, aligning well with the red reference line. Over land, the PV column is upright, extending higher in the vertical,
signifying intense deep convection. Over the ocean, anomalous PV to the north of the features can be observed, especially in the 600–700 hPa range, which coincides with the peak intensity zone of the AEJ. The contours in the Figure distinctly mark the core of the AEJ just north of the wave centers. This lets us suggest that this PV anomaly in the composite can be traced back to PV advection taking place from tropical West Africa to the Atlantic Ocean.

    The ERA-5 modeled precipitation and rainfall estimated by satellite retrievals further match with the interpretation from the
first composite. Both data sets reveal good agreement over the ocean, while significant differences appear over land. Centered around the trough, much less precipitation is observed. Following the argument by Fink and Reiner (2003), squall lines tend to move faster than the trough, and therefore move out of the displayed latitude-pressure cross-section.

    Figure 8a shows box plots depicting the orientation of PV features along the longitude-pressure plane. As described in Sect. 2.6, projecting a 3-D ellipsoid results in an ellipse, which is defined by two main axes. The more vertically oriented axis
is used to calculate the angle of the PV feature relative to the pressure axis. The plot clearly illustrates distinct orientation patterns of PV features depending on their location. Over land, the features exhibit a downshear tilt, while over the ocean, the PV column appears upright, aligning with the composite shown in Fig. 6, validating the structure of the identified ellipsoids. Figure 8b showcases similar box plots for the latitude-pressure plane, which can be visualized by observing the PV features looking from east to west. While the longitude-pressure cross-section is visible in Fig. 6 and has been studied in the literature
(e.g., Russell et al., 2020), the latitude-pressure cross-section (composite in Fig. 7) remains relatively unexplored. Here, a clear tilt pattern is evident as well, which differs between ocean and land. Over the Atlantic Ocean, the PV column tilts to the south, while it appears more upright over land, as also evident in the composite. As explained earlier, we propose that this southern tilt results from PV advection along the AEJ, inducing PV anomalies on the northern flank of the waves at lower levels and creating this asymmetry. Further research could explore the decoupling of PV sources to validate this advection process, distinguishing
advection from diabatically generated PV, similar to Tomassini et al. (2017).

    Figure 9 shows key characteristics of identified PV features within the study area split by longitude, more specifically their mean level and volume. As the AEWs progress from their genesis region towards 0°W, there is a noticeable upward trend in the mean level of the PV features. This upward trend in altitude suggests the evolution of the convective structure associated with the wave, evolving into deep convective structures. Subsequently, beyond 0°W, the mean level gradually decreases until
approximately 40°W, indicating a descent of the diabatic heating centers in the AEWs and a shift towards lower atmospheric

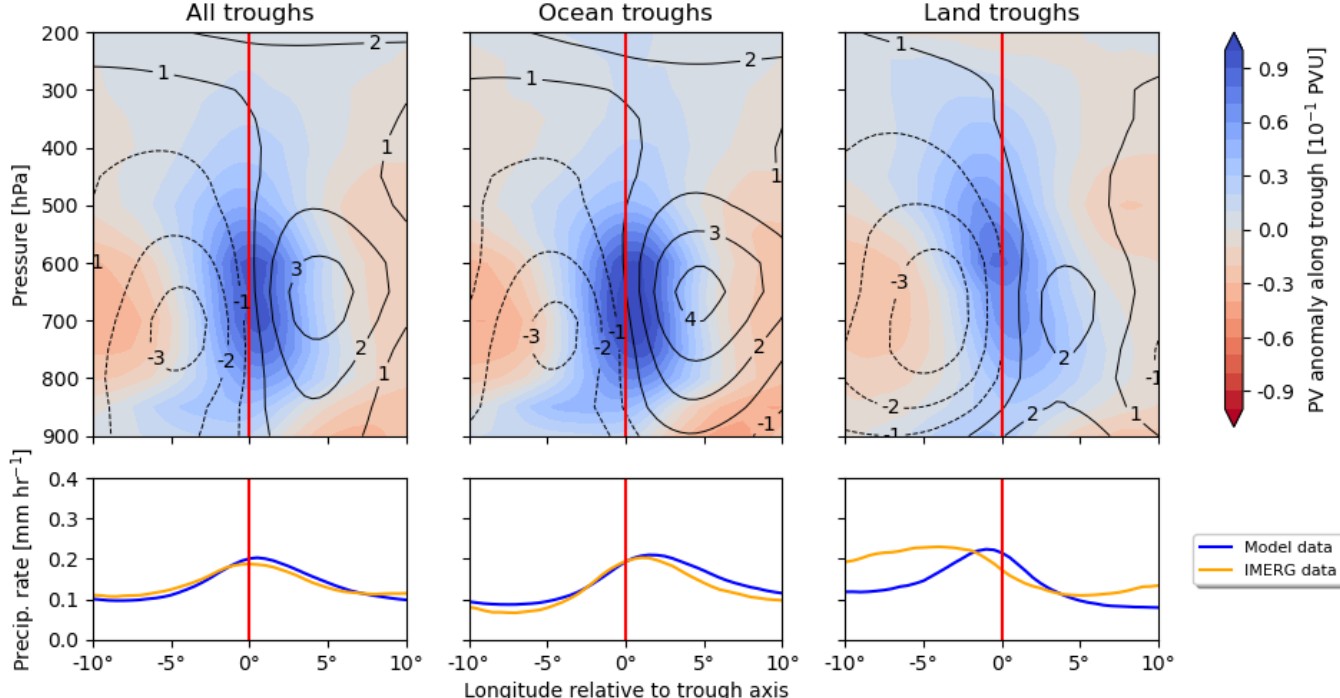

**Figure 6.** Composite of PV anomalies relative to the identified wave troughs along a longitude-pressure cross-section for June–October 2002–2022 based on the ERA-5 reanalysis. The red line indicates the longitude of the wave trough, with regions to the left denoting positions ahead (west) of the trough and those to the right indicating positions behind (east) of the trough. Line contours denote the mean meridional wind around the identified troughs in $ms^{-1}$. Shown are: (a) all identified wave troughs which are part of tracks, (b) focuses on the subset of troughs over the North Atlantic Ocean, and (c) focuses only on wave troughs over West Africa. In the lower panels, 6-hourly centered satellite-estimated (GPM-IMERG) and model-predicted (ERA-5 short-range forecasts) rainfall along this cross-section is visualized.

levels. This is consistent with the composite figures and reflects less deep convection over the ocean. Changes in PV feature volume mirror the evolution of convection and diabatic processes. Peak volumes correspond to the mature phase of convective activity, aligning with the deep convection phase in these regions (Maranan et al., 2018). The orientation of the features depicted by the grey line also confirms the findings from the box-plots in Fig. 8. An abrupt change in orientation is visible across the land-ocean transition zone, which shows the different dynamical behaviors in these contrasting environments.

## 4 Summary & Conclusion

In this study, we have introduced a novel identification and tracking strategy for 3-D PV features within AEWs, allowing for in-depth analysis of their characteristics and statistical properties. Our algorithm builds on a robust identification and tracking of AEW troughs in 2-D at 700 hPa. Identified wave troughs and tracks for the entire ERA-5 reanalysis 1940–2022 are provided,

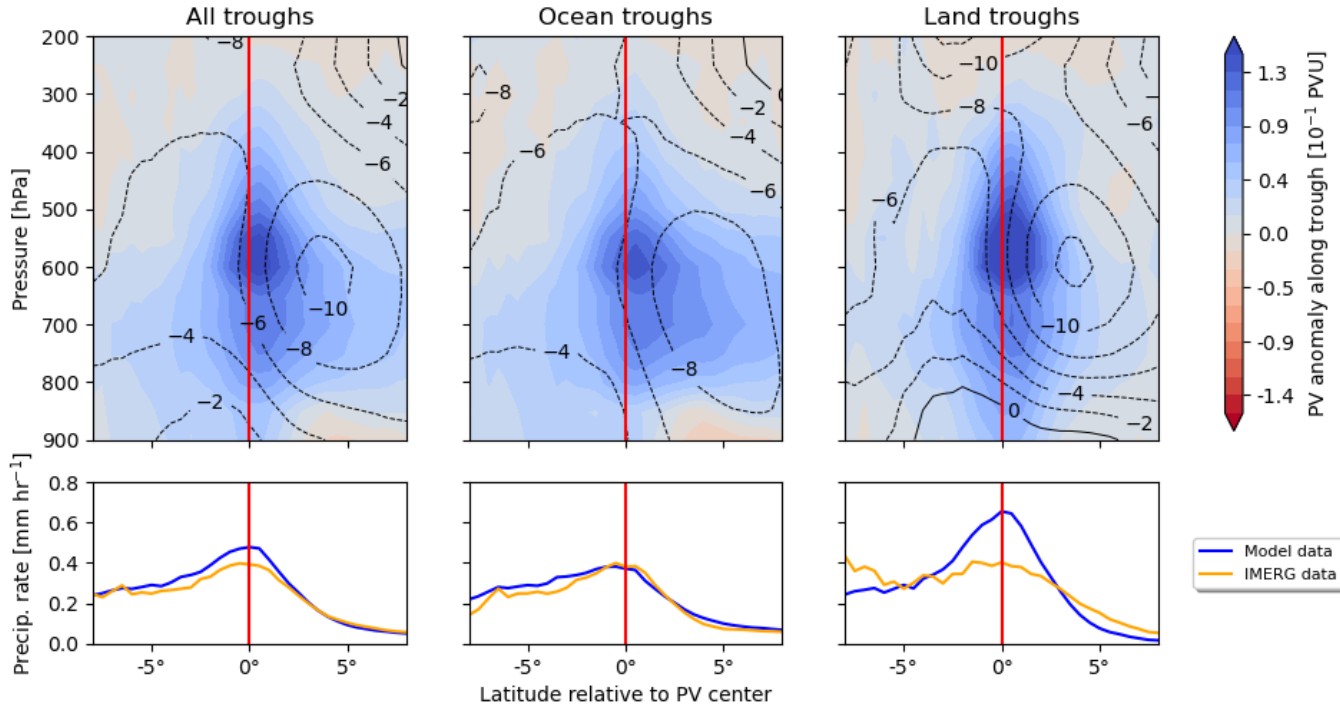

**Figure 7.** Similar to Fig. 6, this composite displays the latitude-pressure cross-section for the same data set, broken down in the same fashion, with the same precipitation data. The contours indicate the mean zonal wind in ms$^{-1}$. The red line denotes the image moment based latitude center of the identified PV anomalies, as outlined in Sect. 2.6.

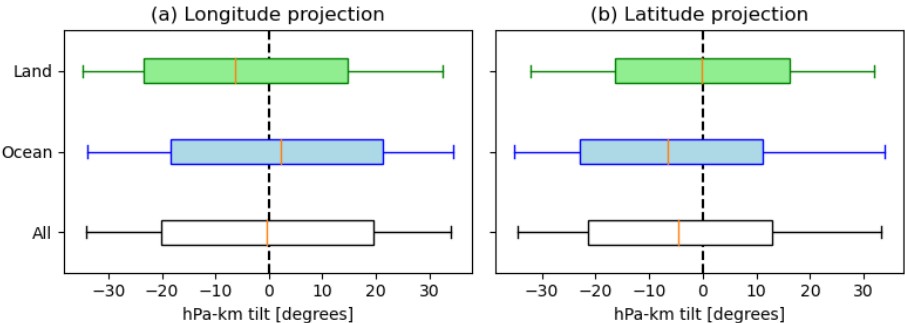

**Figure 8.** Orientation of projected ellipsoids on the (a) longitude-pressure plane and the (b) latitude-pressure plane. Box plots depict the orientation split by location: *Land* contain ellipsoids centered over West Africa, *Ocean* over the North Atlantic Ocean, and *All* indicate all identified features. Positive angles indicate a tilt towards higher longitude (latitude) values with height.

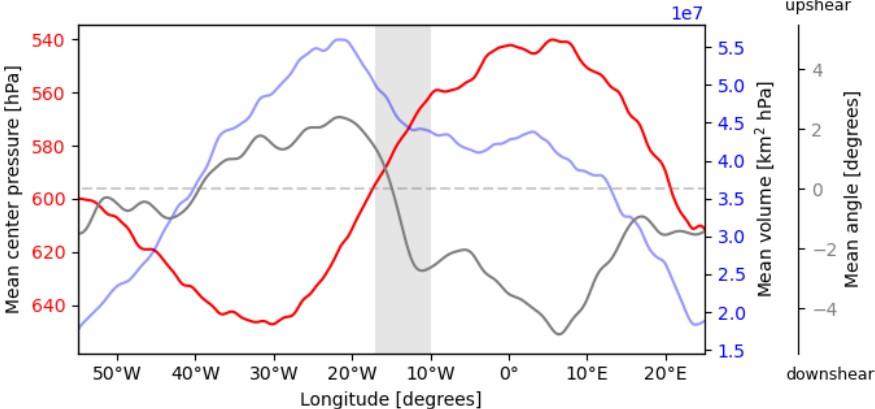

**Figure 9.** Graph showing the mean pressure level (in red), volume (in blue), and angle along the longitude-pressure plane relative to the vertical (in grey) for all identified PV features from June to October, spanning 2002 to 2022, plotted against longitude. The horizontal green line marks the zero angle, indicating a perfectly vertical orientation. The grey area highlights the region where features transition from land to ocean within our study domain.

along with the framework for identifying and tracking meteorological features (see data availability section). The framework offers support for parallel execution and supports output in JSON or Protobuf formats. Furthermore, a near real-time web-page displays wave troughs for ECMWF, GFS, and ICON forecasts. These 2-D wave troughs, combined with the computed wave phase at every point in the study domain, serve as input for the 3-D extension of our strategy, where we extract PV features associated with specific AEWs. Within the identified trough area, we collect and process PV in the column above to create cohesive structures. These PV features are further represented using best-fitting ellipsoids, enabling statistical analyses.

An evaluation of the feature climatology demonstrates the robustness and relevance of our approach. The occurrence, extent, and orientation of the identified PV features closely align with established expert knowledge. By focusing our composite analysis on PV across longitude-pressure and latitude-pressure cross-sections from a trough-centric perspective, we achieved a consistent analysis framework that is not influenced by the seasonal north-south migration of the waves. This climatological review, spanning June to October for the years 2002 to 2022, validates our feature identification approach, particularly highlighting the concurrence between the orientation of PV features in our analysis and existing climatological data.

Additionally, we undertook a comparative analysis of these PV features with estimated (GPM-IMERG) and NWP simulated (ERA-5 short-range forecast) rainfall data. Notable differences in the spatial distribution and intensity of rainfall between the simulated and the estimated data can be observed. These disparities are primarily attributed to variations in the movement of squall lines over tropical West Africa, where satellite estimates show a much faster progression of these lines. A prominent tilt of PV features over tropical West Africa and over the North Atlantic Ocean can be observed both in the composites and in the statistical evaluation of the main axes. An identified southern tilt of these features over the Atlantic suggests PV advection along the AEJ originating from the convective activity over northern Africa. Along with additional feature descriptors, like

mean level and volume of the PV feature, the life-cycle of these waves can be investigated, originating as small disturbance, to

deep moist convection over Africa, and towards less deep convection over the Atlantic Ocean.

     The data generated through our strategy has diverse applications. Besides the generation of feature climatologies, it can be utilized as input for statistical forecasting techniques. Links between PV features and TC activity can be explored by using the features as input to statistical or mixed statistical-dynamical models (Maier-Gerber et al., 2021), as well as for forecasts of tropical rainfall over Western Africa (Vogel et al., 2021). Furthermore, our strategy enables detailed case studies

and facilitates in-depth investigations of AEWs. One particular focus here could be on the relation between the 3-D PV feature evolution and the propensity of an AEW to undergo TC genesis (e.g., Dunkerton et al., 2009; Núñez Ocasio et al., 2020). In all these applications, an important aspect is the sensitivity of the gross characteristics of PV features to different reanalyses or models, because the PV features can be expected to be strongly influenced by the representation of convection. In this context, differences in models with parameterized and explicit convection are considered most worthy of further study. In combination

with state-of-the-art interactive 3-D visualization techniques (Rautenhaus et al., 2018), objective features also open the door for comprehensive case studies.

     In conclusion, our study significantly contributes to the field of AEW research by providing a tool to comprehensively analyze PV features within AEWs, and thus to make the PV perspective more easily applicable to AEWs. The identification and tracking strategy, along with the availability of comprehensive 3-D data, opens up new avenues for studying individual

cases, understanding AEW dynamics, improving forecasts, and gaining a deeper understanding of the role of AEWs in the weather and climate of tropical West Africa and the North Atlantic Ocean.

*Code and data availability.*    The presented method has been implemented into *enstools-feature*, a framework for identification and tracking of meteorological features, which is available and has been archived at Fischer (2023). The computed data set of identified and tracked AEW troughs from the ERA-5 reanalysis is freely available and can be accessed at Fischer et al. (2023a). The data set of identified and tracked PV

features associated with these AEW troughs is freely accessible at Fischer et al. (2023b). The data sets used in this study (ERA-5 reanalysis, GPM-IMERG) are publicly available at the relevant cited sources.

*Author contributions.*    AHF, ES, MRi and MRa supervised and administrated this study. CF designed and implemented the algorithm, performed the climatological analyses, and wrote the publication. AF and MRi contributed meteorological interpretations and analyses. All authors provided feedback on and critical review of the paper.

*Competing interests.*    The authors declare that they have no conflict of interest.

*Acknowledgements.* The research leading to these results has been accomplished within the project C3 "Predictability of tropical and hybrid cyclones over the North Atlantic Ocean" of the Transregional Collaborative Research Center SFB/TRR 165 "Waves to Weather" funded by the German Science Foundation (DFG). The GPM (IMERG) data were provided by the NASA/Goddard Space Flight Center, and archived at the NASA GES DISC.

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
