# Peer review of "An objective identification technique for potential vorticity structures associated with African Easterly Waves"

_Geoscientific Model Development, 2023_

## Referee Comment (RC1)

**An objective identification technique for potential vorticity structures associated with African Easterly Waves**

**by**

**Ch. Fischer et al.**
* * *
**Summary:**

The paper describes a sophisticated methodology how to identify, track and characterize potential vorticity (PV) structures associated with African Easterly Waves (AEWs). The method is split into three distinct steps: (i) AEW troughs are identified and tracked based on curvature vorticity (CV) on 700 hPa; (ii) the AEW troughs are used to associate 3D PV structures to the AEW; and (iii) the complex 3D PV structures are 'fitted' to best-matching ellipsoids to extract geometrical characteristics, e.g., horizontal orientation and vertical tilt. Based on this methodology, AEW PV structures are determined for the whole ERA5 dataset (1940-2023) and monthly occurrence frequencies, AEW-trough-centered PV composites, geometric characteristics are calculated. It is nice that the paper goes beyond a 'simple' description of a novel methodology, and does a first step in climatologically analysing the AEW-associated PV structures. It is, however, also clear that the analysis in the second part of the paper can (and is) only be a first step in this direction, and accordingly the authors also provide some specific ideas what could be done in forthcoming studies.

The paper is well written, the figures are well suited to explain the complex methodology, and the first climatological results are sufficient to trigger/motivate further studies. There are, however, several parts where the text could be a little clearer and more specific and/or the figures be simplified. Some suggestions for improvements, mostly minor, are listed below. If these suggestions are adequately addressed, I can recommend the study to be published in GMD, which seems to be the appropriate journal to describe this new algorithm.

**Major comments:**

**1) Abstract:** The abstract could be somewhat more specific, some of the statements remain rather vague. Further, some of the sentences are rather complicated or are not completely clear. Some examples:

- L3-5: "The dynamics of AEWs can be described in a PV framework and redistribution of PV by latent heat release links PV to the important cloud and rain processes within the wave" → rephrase in easier way.

- L5: What does 'comprehensive' mean in this context?

- L8-9: "These structures are subsequently characterized by low-dimensional descriptors, including spatial information, intensity, and geometric approximations." → This is rather vague, and some more specific details would be helpful. What 'spatial information', 'geometric approximations' are given?

- L10: 'A climatological analysis' → Please specify over which period the climatology is compiled.

- L14-15: 'statistical analysis of main axes agree on an downshear tilt of the PV feature over land, and a tilt to the south over the ocean' → What exactly is meant by 'statistical analysis'? The sentence could also rephrased in a more reader-friendly way.

- L17-19: "The low-dimensional representation of PV features enables the research on further statistical analyses, for example the relationship of these features with tropical cyclogenesis or as predictor for rainfall in the tropics" → Okay, I am not sure whether this outlook, advertisement for further use is appropriate or needed at the end of the abstract. Possibly, the focus is actually on the *low-dimensionality* of the PV-feature respresentation. If so, this should be made more clear to the reader.

**2) Introduction:** The introduction gives a nice overview on AEWs, i.e., about their relevance, characteristics and identification. What remains, however, less clear to me why it is worthwhile/important to study PV structures associated to the AEW. Of course, I fully understand why PV is important in extratropical dynamics and how it facilitates understanding of dynamical processes. This is much less clear in the tropics, where PV dynamics is less established. I would appreciate if the benefit of considering PV in AEW, in addition to more direct measures of diabatic processes (for instance, rainfall or diabatic heating), is discussed in somewhat more detail. From the results I see that the PV disturbances associated with and/or created by AEW display an interesting vertical structure (mean altitude, orientation, vertical tilt), which certainly is interesting. But, dynamically, how important are these PV structures? In particular, since the identified PV structures are not very coherent structures and display a rather complex substructure. A few sentences discussing this aspect would be nice.

3) **Climatology:** It is nice that the new method is applied to the whole ERA5 dataset and thus a climatology is compiled. I wonder whether Section 3 has the best-possible structure. At the  moment, it starts with PV composites in Figure 4 and 5, then proceeds with trough-frequency maps in Figures 6 and 7, before returning in Figure 8 and 9 to some PV-structure characteristics. To me, it would have been more 'natural' to start with the maps of the AEW track frequencies, and then to combine and discuss all aspects of the identified PV structures (composites and geometric characterisation). I assume that the authors wanted to start with the PV composites, as they are the focus of the study. However, if they agree with my storyline argument, they might consider re-arranging the text.

A key result of the study is that the PV structures exhibit a different orientation over ocean and land. This is clearly discernible in the PV composites in Figure 4 and 5, but also in the geometric ellipsoid characterisation in Figure 9. The authors attribute this difference to the enhanced convection over land compared to the ocean, which I think is reasonable. We see, however, also temporal (longitudinal) development over land and over the oceans, and I wonder whether the it would be worthwhile to study not only the land-ocean contrast, but – similar to Figure 9 – to investigate also how the orientation parameters evolve as a function of longitude. In particular, is a tilting already discernible as the AEW troughs and PV structures move from their origin further inland towards the African west coast. I assume that such an analysis would be feasible with little effort, but could provide additional insight (and trigger new research).

In the composite fields, Figure 4 and 5, possibly some additional meteorological fields could be included. Would it, for instance, make sense to show also the wind speed, in particular in the latitude-pressure cross-sections, to see the AEJ together with the PV structures?

**4) Summary & Conclusion:** At some parts, the statmements in the conclusions could be more specific. As an example, L353-355 (These cross-sections…. Feature characteristics) remains rather vague. I would also appreciate if the conclusions are structured in a clearer way, in particular clearly separating results of the study (including the methods) and the potential applications in forthcoming studies. As a side remark, I like the detailed description of potential applications in L364-374, and this kind of PV-motivation might be missing in the introduction (see my point 2 above).

**Specific comments:**

- L29: "The potential vorticity (PV) framework is a fundamental fluid-dynamical concept" → 'concept' reads a little strange; can a 'framework' be a

- L62-65: I wonder whether it would be better to keep the research questions very concise, just the questions, and to move the context to another place. One, possibility would be to move L62-65 to L72, after '...is absolutely vital.'

- L76: The abbreviation 'CV' is introduced here; I would introduce it only later at L104, because at L104 I had to search for it.

- L92: Split the long sentence here: "...2017). The data is available on"

- L96: Is PV calculated on model levels, or based on pressure-level data?

- Section 2.2: The section could be more concise. For instance, the first paragraph gives a brief overview on the algorithm, then the first sentence of the second paragraph seem rather repetitive. I assume, that this sentence wants to introduce that the AEW identification builds on CV *anomalies*, which thus might be highlighted somewhat more.

- L120: "and the advection of CVAs must be equal to 0" → What does this exactly mean? How is the CVA advection determined, and must really be *exactly* zero?

- L126-130: Here, the tracking of the AEWs is described. It is based on a trough-overlap criterion, which at this place of the text is somewhat misleading as the AEW troughs are identified as lines, and thus makes the reader wondering whether really the overlap of the line objects is reasonable. It becomes clear further down in the text, when the forward-projected area for a tAEW trough is introduced and thus the next-step AEW trough have to fall into this area. It would be good not to mislead the reader ion the beginning.

- L133: "AEWs typically propagate with a speed" → "AEWs typically westward propagate with a speed"

- L137-138: "Consequently, … 6-hour difference" → Okay, but quite obvious to me and thus not really necessary.

- L150:152: "Therefore, different sets of tracks can be justified as a solution to this problem. Here, we adhere to the rule to generate tracks with the longest possible life span. This makes it easier to investigate the life cycle of these waves. To achieve that, the graph G is scanned and each node is assigned a time until dissipation" → It is okay to apply the longest-path criterion. However, it would be worthwhile to discuss in one, two sentences alternative approaches, and contrast the approaches with theie advantages and disadvantages.

- Figure 1: The figure nicely illustrates the single steps in the AEW trough identification. Would it make sense to show in panels (a) and (b) wind vectors or the streamfunction instead of streamlines, as the latter can be quite misleading as they do not include information on the wind speed and thus can produce rather artificial structures (whirls) where actually weaks are very weak. Also, the grey lines in panel (a), zero CVA advection, are rather difficult to see; and I wonder whether the streamlines are needed in panels (a) and (b). Finally, the authors might consider whether it would be better (and reasonable) to show in all three panels the same geographical domain.

- L156: "We discard parts of tracks with an average speed of less than 3 ms−1 at any given point in time" → How is the mean speed of the AEW tracks determined. This is not immediately clear as the tracked objects are line-objects.

- L166: "...and is available" → "...and it is available"

- Section 2.4: This section describes the AEW phase computation based on the Hilbert transform. To me, it was somewhat unexpected that the paragraph starts with the wave-associated PV. I see that the authors want to motivate their phase computation with the wave-relative location of convection and diabatic PV production. I wonder, however, whether this is the best place to motivate it, or whether this information fits more nicely into the introduction, hence allowing this section 2.2. to discuss directly the Hilbert tranform.

- L198: "that not near" → "that are not near"

- Figure 2: Are the streamlines really necessary in this figure? It would be better to make the figure less 'noisy'. Further, as also in Figure 1, I would remove all country borders in the maps. They are no needed, as no specific countries are mentioned in the text; removing the border lines would 'de-noise' the figures.

- L219: "removes small outliers while preserving the shape of a structure" → What is exactly meant by 'small outlier'? Is it thus a size/volume filter, i.e., too small PV features are removed? Some further details on the filtering technique might be helpful.

- Figure 3: The color-shading, strictly speaking, does not correspond to height (as mentioned in the caption) but to pressure. Provide also in the figure caption the unit of the shading (hPa).

- Section 2.7: I am not completely sure whether this mini-section is needed, as it is also included in the *Data Availability* section, or what is missing there you be added.

- Figure 6: Here, definitely, the country borders should be removed to make the meteorological fields more prominert; Make also the x-axis labels in the panel 'Occurrence of PV in wavetroughs" consistent to the other panels

- L352: I would remove "and previous analyses within the field"; it is rather unspecific.

- L353: "around identified waves" → "around identified wave troughs"

- L376: remove 'valuable'

- L356: To what is 'Notable differences' referring back? The sentence before addresses a comparison between PV features and observed/simulated rainfall data. Hence, does 'differences' refer to the comparison between PV and rainfall? This is not completely clear to me.

---

## Referee Comment (RC2)

Manuscript: egusphere-2023-218

**An objective identification technique for potential vorticity structures associated with African Easterly Waves**

The manuscript is well-structured, introducing a new identification and tracking approach for African easterly waves (AEWs) that incorporates the 3-D PV structure of the waves. The tracker is promising and provides a new method for analyzing AEWs. It is great to see the authors sharing this tool with the community. Nevertheless, this reviewer has a few major concerns that must be addressed for publication. I encourage the authors to view these comments as an opportunity for improvement.

One significant concern is that, while the authors do include a review of existing trackers, the introduction lacks references and discussion of more recent trackers. This reviewer suggests that the authors expand on the applications of the trackers already mentioned and incorporate additional references to recent trackers along with their applications. Below are a few references the authors should include:

AEW tracker by Lawton: https://doi.org/10.1175/MWR-D-21-0321.1

Lawton's AEW tracker applied: https://doi.org/10.1175/MWR-D-23-0005.1

Tropical easterly wave tracks by M. Hollis: https://doi.org/10.1007/s00382-023-07025-w

Alan Brammer's AEW tracker:  https://doi.org/10.1175/MWR-D-15-0106.1

Brammer's AEW tracker applied:

https://doi.org/10.1175/MWR-D-20-0152.1,  https://doi.org/10.1175/JAS-D-20-0339.1

Monsoon low-pressure system tracker by Hurly and Boss 2015: https://doi.org/10.1002/qj.2447

Another concern is that merely comparing the identified and tracked features to climatology is not sufficient for the proper evaluation and validation of this unique identification and tracking method. This reviewer recommends that the authors incorporate a sensitivity test and compare this identification and tracking scheme to at least one other algorithm or tracker. A simple sensitivity test, where the authors can compare differences in frequency, intensity, splits, mergers, initiation, and termination, would be highly valuable for the AEW-analyzing community and would support the validity of the proposed technique. The authors do not necessarily need to run the algorithms for the entire climatology, but adding a section that intercompares them for a small period would certainly be insightful. This type of analysis aligns with current trends in the community, where it is becoming more common to evaluate multiple object-based tools. For reference, see Prein et al. 2024 for an MCS intercomparison study: https://doi.org/10.22541/essoar.169841723.36785590/v1. The authors have a unique opportunity here to compare their results to another tracking algorithm.

Given the intrinsic relationship between PV structures and convection/diabatic heating, this reviewer anticipated the inclusion of figures related to convection, such as diabatic heating profiles or Outgoing Longwave Radiation (OLR). It would enhance the reader's understanding of this relationship and its significance if the authors incorporated such an analysis in climatology.

Furthermore, it would be beneficial for the authors to include a discussion (aside from what is already presented in the conclusions) regarding the tracker and its potential role in distinguishing between developing and non-developing African Easterly Waves (AEWs).

**Specific comments:**

Line 23: AEWs are known to also initiate over high topography, especially over eastern Africa (and thus, the Ethiopian Highlands (Hamilton et al., 2020, Rajarsee et al., 2023 as well as others). Moreover, AEWs that initiate over eastern Africa are more likely to become TC in the eastern Atlantic (i.e., Núñez Ocasio et al., 2021). Please include these references and other relevant references.

Line 27: AEWs are also related to the West African offshore rainfall maximum (i.e., Hamilton et al. 2017).

Line 53: Please rephrase "theoretical perspective" as the other references also incorporate theories in their analysis. Dunkerton et al., 2009 proposed a simpler 'geometrical', forecaster-friendly idea.

Tracking: It is unique that the authors incorporate MCS tracking methods into the tracking of AEWs. Both overlapping (Houze, Evans, and Shemo, PyFLEXTRKR by Z. Feng, TAMS by Núñez Ocasio, and MOAAP by Andreas Prein) and graph theory (Kim Whitehall) are used to track MCSs. Please make a point of this and include references.

Line 150: What features do the authors refer to? It is more likely for convection or PV features to split and merge than the actual AEW splitting and merging. Please clarify.

Line 206: And how do you account for these artifacts?

Figure 3: Is shading height or PVU contours? Please clarify and add units.

Line 227: As well as thermodynamic constraints (Núñez Ocasio and Rios Berrios 2023). Please add.

Lines 360: The AEJ was not studied in detail and so it is a speculation. Remove or add such analysis.

---

## Author Response (AR1)

**RESPONSE TO REVIEWERS**
**An objective identification technique for potential vorticity structures associated with African Easterly Waves**
**gmd-2023-218**

We would like to thank both reviewers and the editor for their time spent on reviewing our manuscript and their thoughtful comments that helped to improve the article. Below, we provide detailed point-by-point replies (blue font color) to each comment. Technically, citations of text passages are in italics. Text changes are highlighted in yellow. Line numbers refer to the resubmitted manuscript.

**Reviewer # 1:**

Major comments:

The reviewer states that some of the statements in the abstract could be less vague and more specific, and other statements are not completely clear. Specifically, the reviewer mentioned:

- L3-5: "The dynamics of AEWs can be described in a PV framework and redistribution of PV by latent heat release links PV to the important cloud and rain processes within the wave" → rephrase in easier way.
Rephrased as outlined below.

- L5: What does 'comprehensive' mean in this context?
We decided on removing this specific word in this context.

- L8-9: "These structures are subsequently characterized by low-dimensional descriptors, including spatial information, intensity, and geometric approximations." → This is rather vague, and some more specific details would be helpful. What 'spatial information', 'geometric approximations' are given?
Rephrased as outlined below. To simplify, we now mention some specific key characteristics of a feature: its location, intensity, and orientation.

- L10: 'A climatological analysis' → Please specify over which period the climatology is compiled
Added as outlined below.

- L14-15: 'statistical analysis of main axes agree on an downshear tilt of the PV feature over land, and a tilt to the south over the ocean' → What exactly is meant by 'statistical analysis'? The sentence could also rephrased in a more reader-friendly way
We suggest a rather simple version below by omitting detailed evaluations.

- L17-19: "The low-dimensional representation of PV features enables the research on further statistical analyses, for example the relationship of these features with tropical cyclogenesis or as predictor for rainfall in the tropics" → Okay, I am not sure whether this outlook, advertisement for further use is appropriate or needed at the end of the abstract. Possibly, the focus is actually on the *low-dimensionality* of the PV-feature respresentation. If so, this should be made more clear to the reader

We rephrase this section to be more centered on the low-dimensionality aspect.

The abstract has been revised as follows to address the concerns and suggestions:

**Revised abstract:**
*"Tropical Africa and the North Atlantic Ocean are significantly influenced by African Easterly Waves (AEWs), which play a fundamental role in tropical rainfall and cyclogenesis in that region. The dynamics of AEWs can be described in a potential vorticity (PV) framework. The important impact of latent heat release by cloud processes is captured in this framework by the diabatic generation of PV anomalies. This paper introduces an innovative approach for  the identification and tracking of PV structures within AEWs.*

**

**

*By employing AEW tracking and computing the wave phase of each point within the AEW domain using a Hilbert transform, we are able to effectively identify and collect 3-D PV structures associated with specific AEWs. To facilitate a climatological analysis, here performed over the months June to October from 2002 to 2022, these structures are subsequently characterized by low-dimensional descriptors, including their location, intensity, and orientation.*

*Our climatological analysis reveals the seasonal evolution and the structural attributes of PV anomalies within AEWs over the study domain. PV feature locations closely align with the African Easterly Jet's latitudinal shift during the summer season. Analysis of the mean pressure level of the 3-D PV structures shows a remarkable shift during their life cycle, indicating deep moist convection characteristics over land, and more shallow convection characteristics over the ocean. On average, PV features identified within AEW troughs tilt downshear over land and equatorward over the ocean.  The trough-centered analysis reveals distinct differences between satellite-estimated and model-predicted rainfall. Agreement between the results of a more traditional composite analysis and our new feature analysis provides confidence in our feature approach as a novel diagnostic tool. The feature framework provides a low-dimensional representation of AEWs' PV structure, which facilitates future statistical analyses of the relation of this structure to, e.g., tropical cyclogenesis or predictability of tropical rainfall."*

In the Introduction, it remains unclear to the reviewer why it is worthwhile to study PV in the tropics, and especially associated with AEWs, and to what extent these PV structures are dynamically important. The reviewer notes:
What remains, however, less clear to me why it is worthwhile/important to study PV structures associated to the AEW. Of course, I fully understand why PV is important in extratropical dynamics and how it facilitates understanding of dynamical processes. This is

much less clear in the tropics, where PV dynamics is less established. I would appreciate if the benefit of considering PV in AEW, in addition to more direct measures of diabatic processes (for instance, rainfall or diabatic heating), is discussed in somewhat more detail. From the results I see that the PV disturbances associated with and/or created by AEW display an interesting vertical structure (mean altitude, orientation, vertical tilt), which certainly is interesting. But, dynamically, how important are these PV structures? In particular, since the identified PV structures are not very coherent structures and display a rather complex substructure.

We appreciate the reviewer's insightful comments and agree that the importance of studying PV structures associated with AEWs in tropical dynamics could be more explicitly articulated. We recognize that while PV dynamics is well-established in extratropical regions, its significance in the tropics, particularly in relation to AEWs, warrants further clarification. Therefore, we revise this section as followed (Lines 42ff):

"… *A feature-based PV perspective of AEWs thus faces challenges that demand a more detailed investigation.*

*There exists a substantial body of research on AEWs, especially encompassing their dynamical interaction with the environment and their relationship with TCs. The dry dynamics of AEWs can be understood in terms of downstream propagation along the African Easterly Jet (AEJ) from an upstream wave source (Thorncroft et al., 2008), with (small) amplification by baroclinic and barotropic growth (Hall et al., 2006). As in the midlatitudes, these processes can be described from the PV perspective. More important for AEW amplification is latent heat release associated with embedded convection (Berry and Thorncroft 2005; Thorncroft et al., 2008). From a PV perspective, this amplification is seen as the diabatic generation of PV anomalies. Besides amplitude, the diabatically generated PV anomalies signify modification of AEW structure. Tomassini et al. (2017) investigated in detail the contributions of different parameterization schemes to diabatically modified PV in an operational numerical model. Russell et al. (2017, 2020) give further insight into the structure and sources of PV in AEWs, including the role of moist convection and its coupling with the background wave environment. Essentially, the diabatically generated PV anomalies encapsulate the impact of moist processes on AEW intensity and structure that outlasts a period of active convection.*

*In terms of AEW predictability, there is strong indication that …"*

Newly added references:
Hall, N. M. J., Kiladis, G. N., & Thorncroft, C. D. (2006). Three-Dimensional Structure and Dynamics of African Easterly Waves. Part II: Dynamical Modes. Journal of the Atmospheric Sciences, 63(9), 2231-2245. https://doi.org/10.1175/JAS3742.1

In the climatology section, the reviewer suggests to change the general structure and order of the figures. Currently, we start with a composite of PV structures along the identified wave troughs, continuing with a general wave track occurrence schematic, and then going into detail on the structures and characteristics of the 3-D PV features. The reviewer suggests to start with the track occurrences, before going and staying with the PV features.

We thank the reviewer for this input and fully agree. Especially since we are introducing the 2-D wave tracking first, it also makes sense to start with their analysis before getting into the more sophisticated ones. This reordering does not include any substantial text changes. The climatology chapter now starts as follows (Lines 293ff):

*To assess the data set generated by the identification algorithm and provide an overview of the features, a climatological analysis is conducted.* ==Figure 4*== *presents a climatology depicting the occurrence of wave troughs and geometrical representations. [...]*

New figures 4 and 5 are interpreted here with no text changes.

*[...] PV activity over West Africa retreats southward as well in tandem with the monsoon's withdrawal.*
==*As 3-D reference, Figure 6* shows a composite of the PV*== *anomalies relative to the identified wave troughs, [...]*

continuing with the 3-D interpretation. Related, the figures are reordered as followed:
* new Figure 4: former Figure 6
* new Figure 5: former Figure 7
* new Figure 6: former Figure 4
* new Figure 7: former Figure 5

A key result of the study is that the PV structures exhibit a different orientation over ocean and land. This is clearly discernible in the PV composites in Figure 4 and 5, but also in the geometric ellipsoid characterisation in Figure 9. The authors attribute this difference to the enhanced convection over land compared to the ocean, which I think is reasonable. We see, however, also temporal (longitudinal) development over land and over the oceans, and I wonder whether the it would be worthwhile to study not only the land-ocean contrast, but – similar to Figure 9 – to investigate also how the orientation parameters evolve as a function of longitude. In particular, is a tilting already discernible as the AEW troughs and PV structures move from their origin further inland towards the African west coast. I assume that such an analysis would be feasible with little effort, but could provide additional insight (and trigger new research).

We thank the reviewer for this comment and agree that adding such information and an analysis would be a great idea that might also trigger new research. We also computed the angle of the PV features by longitude, similar to the other metrics, and added it to the line plot:

[Figure]

Clearly, the signal is distinct and fits nicely with the shown composite data. In the transition area between land and ocean, the angle (along with other properties) do change significantly. We also added a vertical line indicating roughly the border between ocean and land in the domain, as well a horizontal line indicating the upright position of the feature. To remove clutter, we removed the other horizontal lines. The revised caption of the above revised Figure 9 now says:

*"Graph showing the mean pressure level (in red), volume (in blue), and angle along the longitude-pressure plane relative to the vertical (in grey) for all identified PV features from June to October, spanning 2002 to 2022, plotted against longitude. The horizontal green line marks the zero angle, indicating a perfectly vertical orientation. The grey area highlights the region where features transition from land to ocean within our study domain."*

Furthermore, with this addition, we changed the order of the figures to first show the boxplots for the orientation, and then, as a more detailed view of the evolution of the tilt the line plot. The text for the boxplot figure does not change by this. However, we add a few sentences commenting on the evolution of the orientation (Lines 368ff):

*"[...] Peak volumes correspond to the mature phase of convective activity, aligning with the deep convection phase in these regions (Maranan et al., 2018). The orientation of the features depicted by the grey line also confirms the findings from the box-plots in Fig. 8. An abrupt change in orientation is visible across the land-ocean transition zone, which shows the different dynamical behaviors in these contrasting environments."*

In the composite fields, Figure 4 and 5, possibly some additional meteorological fields could be included. Would it, for instance, make sense to show also the wind speed, in particular in the latitude-pressure cross-sections, to see the AEJ together with the PV structures?

Other meteorological fields would be a great addition, both to showcase the AEW-centric view of the composite, and to further understand and interpret the results. We generated the wave trough centric U and V wind, and added the U wind to the latitude composite, and the

V wind to the longitude composite. The results are indeed very useful and show a clear signal. The figures now look as followed:

[Figure]

The captions got revised to:
*"[...] The red line indicates the longitude of the wave trough, with regions to the left denoting positions ahead (west) of the trough and those to the right indicating positions behind (east) of the trough. ==Line contours denote the mean meridional wind around the identified troughs in ms-1.== Shown are: (a) [...]"*
and
*"Similar to Fig. 4, this composite displays the latitude-pressure cross-section for the same data set, broken down in the same fashion, with the same precipitation data. ==The contours indicate the mean zonal wind in ms-1.== The red line denotes the image moment based latitude center of the identified PV anomalies, as outlined in Sect. 2.6."*

Furthermore, when referencing the figures from the climatology chapter, we add a short remark on this addition (Lines 322ff):

"Over land, the PV column exhibits a noticeable downshear tilt (the lower tropospheric shear vector points westward due to the AEJ), whereas over the ocean, the PV column appears upright in relation to this cross-section. Centered around the wave trough, a clear dipole structure in the meridional wind can be observed, with a maximum around 650-700 hPa, falling in line with the maximum intensity of the AEJ (e.g, Burpee, 1972). Moreover, over land high PV values extend higher into the troposphere than over ocean. [...]"

"[...] Over land, the PV column is upright, extending higher in the vertical, signifying intense deep convection. Over the ocean, anomalous PV to the north of the features can be observed, especially in the 600-700 hPa range, which coincides with the peak intensity zone of the AEJ. The contours in the Figure distinctly mark the core of the AEJ just north of the wave centers. This lets us suggest that the PV anomaly in the composite can be traced back to PV advection taking place from tropical West Africa to the Atlantic Ocean."

At some parts, the statmements in the conclusions could be more specific. As an example, L353-355 (These cross-sections.... Feature characteristics) remains rather vague. I would also appreciate if the conclusions are structured in a clearer way, in particular clearly separating results of the study (including the methods) and the potential applications in forthcoming studies. As a side remark, I like the detailed description of potential applications in L364-374, and this kind of PV-motivation might be missing in the introduction (see my point 2 above).

The remark on the structure of the summary is a bit unclear to us. While the first paragraph gives a brief overview of the method, the second summarizes the findings of the climatological analysis, and the third paragraph lists multiple applications of the strategy. We consider the summary therefore well-structured and separated regarding the given remark. If the reviewer can give us more information or a more detailed hint on how the structure can be improved in the context of the given summary, we are happy to discuss this further.

However, we agree that the specific section of the summary remains a bit vague. While trying to keep the summary concise, we added a few sentences on the significance of these cross sections analyzed. The second paragraph of the summary reads now as (Lines 381ff):

"An evaluation of the feature climatology demonstrates the robustness and relevance of our approach. The occurrence, extent, and orientation of the identified PV features closely align with established expert knowledge and previous analyses within the field. By focusing our composite analysis on PV across longitude-pressure and latitude-pressure cross-sections from a trough-centric perspective, we achieved a consistent analysis framework that is not influenced by the seasonal north-south migration of the waves. This climatological review, spanning June to October for the years 2002 to 2022, validates our feature identification approach, particularly highlighting the concurrence between the orientation of PV features in our analysis and existing climatological data.

*Additionally, we undertook a comparative analysis of these PV features with estimated (GPM-IMERG) and NWP simulated (ERA-5 short-range forecast) rainfall data. Notable differences in the spatial distribution and intensity of rainfall can be observed. [...]"*

Specific comments:

We address the specific comments by Reviewer #1 as follows:

- L29: "The potential vorticity (PV) framework is a fundamental fluid-dynamical concept" → 'concept' reads a little strange; can a 'framework' be a

We clarified the term "concept" as a conceptual model, which should make it more clear to the authors (Lines 30ff).
*The potential vorticity (PV) framework is a fundamental fluid-dynamical conceptual model widely utilized in extratropical meteorology, including the understanding of barotropic and baroclinic instabilities [...]*

- L62-65: I wonder whether it would be better to keep the research questions very concise, just the questions, and to move the context to another place. One, possibility would be to move L62-65 to L72, after '...is absolutely vital.'

We agree with the reviewer and moved the sentence, which are not directly linked to the research question, to the paragraph after the research questions. Without quoting the change directly, we moved the sentence *"Objective identification techniques [...] Hengstebeck et al., 2011)."* to line 75 after *"[...] is absolutely vital."*. This is reflected in the track changes file.

- L76: The abbreviation 'CV' is introduced here; I would introduce it only later at L104, because at L104 I had to search for it.

Since the term CV is used extensively in Section 2, but only once in the Introduction, we also deem an introduction of this term there reasonable. Line 81 now does not include the abbreviation anymore *"[...] used the advection of curvature vorticity as primary measure [...]"*, and line 118 now reads *"[...] by applying curvature vorticity (CV) instead of absolute values of relative vorticity [...]"*.

- L92: Split the long sentence here: "...2017). The data is available on"

We split the sentence as follows to improve readability (Lines 103ff):
*"For the identification of AEWs and the corresponding PV features, as well as for the analyses in this study, we utilize data from the global ERA-5 reanalysis (Hersbach et al., 2020). Our analysis focuses on the period from June to October to align with the West African Monsoon season, as detailed by Fink et al. (2017). The selected data is provided on a regular grid with a grid-point spacing of 0.5° in both latitude and longitude, with a temporal resolution of 6 hours."*

- L96: Is PV calculated on model levels, or based on pressure-level data?

The variables present in the ERA5 data set are all computed on all 137 available model levels, and at the end interpolated onto pressure levels. We do not compute the PV on our own. We added a short remark to include this fact (Lines 110ff):

*"The PV analysis is conducted on 16 pressure levels between 200 and 900 hPa, with intervals of 50 hPa. ==The PV data on pressure levels originates from the ERA-5 archive."==*

- Section 2.2: The section could be more concise. For instance, the first paragraph gives a brief overview on the algorithm, then the first sentence of the second paragraph seem rather repetitive. I assume, that this sentence wants to introduce that the AEW identification builds on CV *anomalies*, which thus might be highlighted somewhat more.

We rephrased the paragraphs in question to be more concise in general, by moving sections to the first paragraphs and removing redundant information. So, the first paragraphs gives a quick motivation of the strategy and the changes we made, then the second paragraph goes over the actual methodology (Lines 116ff):

*"To robustly identify PV features within AEWs, we build on the method by Belanger et al. (2016) to firstly identify AEW troughs on 700 hPa. Their method is an improvement over previous work (Thorncroft et al., 2001) by ==applying CV anomalies instead of absolute values of relative vorticity and by ensuring that waves are westward propagating, more closely aligning with the characteristics of AEWs. We have improved the tracking of the waves to achieve more robust and consistent tracks.== Additionally, we introduce a data structure to facilitate the analysis of the tracks, including the identification of split and merge events, computation of average wave speeds, and other parameters.*

*==== A climatology of CV is computed for each month and each time of day to take into account both seasonal and diurnal effects. [...]"*

- L120: "and the advection of CVAs must be equal to 0" → What does this exactly mean? How is the CVA advection determined, and must really be *exactly* zero?

That is an important remark and missing in the current manuscript. The advection is calculated using the advection operator, thus CV * ∇, thus in a perfectly easterly motion, it collapses to CV * d/dx and becomes zero on maxima and minima along a line of constant latitude. However, clearly from a practical standpoint on discrete grids, the advection will basically never get *exactly* zero. But if one grid cell has a negative advection, and the neighboring cell a positive one, there has to be a zero in between. We are identifying these in-between points by bilinear interpolation between the cells (Marching Squares algorithm), and connecting the zero points (which therefore are not necessarily aligned with the discrete grid) them to line segments making up the wave trough. Therefore, this is not a mask in the context of how it was introduced in this section. Therefore, we rephrase this section as follows (Lines 128ff):

*"Following Belanger et al. (2016), the data is smoothed twice using a 9-point local smoother, which retains only the synoptic-scale easterly wave structure. ==Then, the zeros of the CVA advection are determined, which resemble the lines where the sign of the advection changes. These are the troughs and ridges in the CVA field. Given the grid's discrete nature,==*

*a cell wise computation would almost never find zero-values. Hence, we employ the Marching Squares algorithm to interpolate line segments between grid cells that approximate these zero-lines. These segments are then merged to form continuous lines. In Fig. 1a, the grey lines indicate zeros of CVA advection, which collocate with troughs and ridges.*

*For filtering purposes, following Belanger et al. (2016), two masks are applied to the identified lines:*
*– the zonal wind must be u < +2.5ms−1,*
*– and the CVA must be over the 66-th percentile of the entire reanalysis.*
*Furthermore, we consult second derivative of the CVA to extract only troughs in the data set, masking out the ridges. The red lines in Fig. 1a are the result of applying the other masks and filters to the data set. This results in the identified wave troughs."*

- L126-130: Here, the tracking of the AEWs is described. It is based on a trough-overlap criterion, which at this place of the text is somewhat misleading as the AEW troughs are identified as lines, and thus makes the reader wondering whether really the overlap of the line objects is reasonable. It becomes clear further down in the text, when the forward-projected area for a tAEW trough is introduced and thus the next-step AEW trough have to fall into this area. It would be good not to mislead the reader ion the beginning.

We agree that this could be misleading indeed when first mentioning an overlap heuristic for identified lines, but not introducing the forward-projected areas. Therefore, we restructured this section to already state in the beginning that we are looking at forward-projected areas. Furthermore, based on a comment by Reviewer #2, we added references to other feature tracking strategies using a similar concept. After addressing both concerns, this section now reads as followed (Lines 149ff):

*"To form tracks from the identified individual wave troughs, we employ an overlap approach. Overlap tracking has proven to be a robust tracking technique in meteorological applications, such as tracking of Mesoscale Convective Systems (Núñez Ocasio et al, 2020; Feng et al. 2023; Prein et al., 2023) and general purpose feature extraction (Ullrich et al., 2021). However, since our identified wave troughs are represented as line strings, they don't directly lend themselves to traditional overlap tracking methods.*

*To address this, we create area features by predicting the future positions of each trough for upcoming time steps, t+Δt and t+2Δt, with Δt set to 6 hours. This prediction uses an anticipated propagation speed to define a polygonal area that represents where the trough is expected to be. The presence of overlap between these predicted polygonal areas and the actual locations of wave troughs at future time steps facilitates the tracking. The range of expected propagation speeds, which we set from u_max=-15ms-1 to u_min=+2ms-1, defines the size and shape of these polygons, as illustrated in Fig. 1b."*

*While AEWs typically propagate with a speed [...]. Consequently, when comparing wave troughs with a time difference of 12 hours, the resulting polygon covers twice the area compared to a 6-hour difference.*

*With this approach, wave tracks can persist even when a trough is detected at t and t+2Δt, but not at t+Δt. This happens, when one of the above-mentioned identification masks is not*

*fulfilled for this time step, but for the ones before and after. Consequently, maintaining longer continuous wave tracks, rather than multiple short tracks, enhances the quality of the tracking results."*

Newly added references:
Feng, Z., Hardin, J., Barnes, H. C., Li, J., Leung, L. R., Varble, A., & Zhang, Z. (2023). PyFLEXTRKR: a flexible feature tracking Python software for convective cloud analysis. Geoscientific Model Development, 16(10), 2753-2776.
Prein, A. F., Mooney, P. A., & Done, J. M. (2023). The multi-scale interactions of atmospheric phenomenon in mean and extreme precipitation. Earth's Future, 11, e2023EF003534. https://doi.org/10.1029/2023EF003534
Ullrich, P. A., Zarzycki, C. M., McClenny, E. E., Pinheiro, M. C., Stansfield, A. M., and Reed, K. A.: TempestExtremes v2.1: a community framework for feature detection, tracking, and analysis in large datasets, Geosci. Model Dev., 14, 5023–5048, https://doi.org/10.5194/gmd-14-5023-2021, 2021.

- L133: "AEWs typically propagate with a speed" → "AEWs typically westward propagate with a speed"

*"While AEWs typically propagate westward with a speed between [...]"*

- L137-138: "Consequently, … 6-hour difference" → Okay, but quite obvious to me and thus not really necessary.

We agree that this is very clear within the context and from the figure and does not need to be mentioned here. So we removed the sentence (Line 162):

*[...] by multiplying u_min and u_max with the elapsed time. *

- L150:152: "Therefore, different sets of tracks can be justified as a solution to this problem. Here, we adhere to the rule to generate tracks with the longest possible life span. This makes it easier to investigate the life cycle of these waves. To achieve that, the graph G is scanned and each node is assigned a time until dissipation" → It is okay to apply the longestpath criterion. However, it would be worthwhile to discuss in one, two sentences alternative approaches, and contrast the approaches with theie advantages and disadvantages.

We thank the reviewer for this remark and we actually also considered some alternatives, not specific to this application, but more generic in a feature tracking environment. The supplementary framework code actually contains different implementations of feature tracking. While this discussion would be too extensive in the context of the paper, we agree that a very short summary of the discussion is adequate here (Lines 181ff):

*"Therefore, different sets of tracks can be justified as a solution to this problem. Here, we adhere to the rule to generate tracks with the longest possible lifespan. This makes it easier to investigate the life cycle of these waves. Alternative heuristics, such as extracting connected sub-graphs or initiating new tracks at every split and merge event, were considered. Extracting connected sub-graphs could reduce the number of tracks and inherently capture split and merge events within a single track. However, this approach is*

*sensitive to inaccuracies in feature identification and overlooks the origins of merged features, connecting too many features into one. On the other hand, creating new tracks at each split or merge event significantly increases the number of tracks, making life span analyses unfeasible.*

- Figure 1: The figure nicely illustrates the single steps in the AEW trough identification. Would it make sense to show in panels (a) and (b) wind vectors or the streamfunction instead of streamlines, as the latter can be quite misleading as they do not include information on the wind speed and thus can produce rather artificial structures (whirls) where actually weaks are very weak. Also, the grey lines in panel (a), zero CVA advection, are rather difficult to see; and I wonder whether the streamlines are needed in panels (a) and (b). Finally, the authors might consider whether it would be better (and reasonable) to show in all three panels the same geographical domain.

Upon reflection, we agree that panel (a) may have been overly complex, potentially obscuring the key elements of the key messages at first glance. We experimented with incorporating wind barbs to address the suggestion of depicting wind vectors for a clearer representation. However,  including wind barbs did not significantly enhance the figure's readability. In light of this, we have opted to remove the wind information entirely from panel (a) while retaining the streamlines in panel (b), where they contribute to a more coherent visual narrative. This decision also eliminates redundant information, given that panels (a) and (b) represent the same time step. We also improve the visibility of the grey lines by removing the country borders.
Regarding the suggestion to unify the geographical domain in all panels for consistency, we did consider it. However, our tests showed that doing so would make some details too small to see clearly, because we're focusing on different things in each panel. We believe keeping the original sizes of each panel makes it easier to see and understand these details. We hope our decision to stick with the current presentation of the figure, despite the differences in zoom levels, makes sense and addresses your concerns.

Here is the revised figure, and a version of panel (a) using barbs:

[Figure]

The revised caption now says:

"*Illustration of the identification and tracking process for AEWs using a polygonal search approach. (a) shows the*  *CV anomalies and the precipitation rate according to the GPM IMERG data set at 13 September 2022, 00 UTC. Grey lines indicate the zeros of CVA advection, and the red lines show the*  *extracted wave troughs based on the filters introduced in Sect. 2.2. In (b), two identified wave troughs are shown at 00 UTC (red, as in (a)), 06 UTC (orange), and 12 UTC (yellow). Red boxes sketch the polygonal search areas computed for both wave troughs, initiated at 00 UTC. The left wave trough is compared with the +12 h wave trough, while the right wave trough is compared with the +6 h wave trough. The 700 hPa streamlines*

*of the bandpass-filtered wind at 00 UTC are depicted as blue lines. In panel (c), an entire AEW track is depicted, demonstrating the continuity of the tracking process. The three wave troughs from (b) are highlighted."*

- L156: "We discard parts of tracks with an average speed of less than 3 ms−1 at any given point in time" → How is the mean speed of the AEW tracks determined. This is not immediately clear as the tracked objects are line-objects.

We agree that this should be further specified in the manuscript. The previous sentence should give a hint, however, as it currently reads, it can be misleading on how and for what troughs a speed is computed. Therefore, we revised that paragraph as follows (Lines 190ff):

*"Subsequently, the extracted tracks $T^i$ can be individually analyzed and filtered. For each identified wave trough being part of a track, we compute its current speed as the average speed over a 2-day window centered around the trough's time step along this track. For this purpose, the position of a wave trough is defined by the center of its bounding box. We discard parts of tracks with an average speed of less than 3 ms-1 at any given point in time. This removes stationary features [...]."*

- L166: "...and is available" → "...and it is available"

We rephrased the sentence as a whole to the following (Lines 202ff):
*"The computed AEW trough data set, spanning the entire ERA-5 reanalysis period from 1940 to 2022, is available as detailed in the data availability section."*

- Section 2.4: This section describes the AEW phase computation based on the Hilbert transform. To me, it was somewhat unexpected that the paragraph starts with the wave-associated PV. I see that the authors want to motivate their phase computation with the wave-relative location of convection and diabatic PV production. I wonder, however, whether this is the best place to motivate it, or whether this information fits more nicely into the introduction, hence allowing this section 2.2. to discuss directly the Hilbert tranform.

Trying to keep the introduction a bit shorter and concise, we decided in the original manuscript to leave this paragraph in the methodology section. However, we share the concern that it would be still more reasonable to move the meteorological motivation to the introduction to keep the methodology more clear. Therefore, we added a paragraph in the introduction starting Line 83 as follows:

*"To enhance our analysis of areas with high PV associated with AEWs, we incorporate phase filtering through the Hilbert transform. This technique is essential because the geographic location of an AEW trough often does not match the regions with the highest PV anomalies, due to the complex interactions between wave movements and diabatic PV influenced by convection. According to Shapiro (1978) and Tomassini et al. (2017), convection within AEWs typically begins ahead, or west, of the trough over Africa, where atmospheric conditions are most conducive to convection (Reed et al., 1977; Fink and Reiner, 2003). Conversely, over the ocean, convection generally occurs closer to, or slightly east of, the trough (Riehl, 1954). Since convection can potentially take place across the*

*entire trough area, accurately assigning the current phase to every point in the domain becomes crucial for filtering PV signals in AEWs."*

And shortened the first paragraph in Section 2.4 (from line 206) to:

*"Although an AEW trough defines the line of maximum CV, its location does not necessarily coincide with the expected region of maximum PV anomalies, which could occur in the entire trough phase. This discrepancy underscores the need to compute the wave phase for each point across the domain. Looking at the meridional wind component v in the background flow, the trough area can be assigned a phase between -pi to 0, and the ridge a phase between 0 and pi. However, it is not straightforward to extract the phase information from a real-valued field, since all frequencies in the bandpass-filtered range of 2--8 days can contribute to it.*

*Zimin et al. (2003) faced a similar problem [...]"*

- L198: "that not near" → "that are not near"

Fixed the error as proposed to (Lines 220-221):
*"To remove phase signals in the trough area (-pi … 0) that are not near any AEW trough [...]"*

- Figure 2: Are the streamlines really necessary in this figure? It would be better to make the figure less 'noisy'. Further, as also in Figure 1, I would remove all country borders in the maps. They are no needed, as no specific countries are mentioned in the text; removing the border lines would 'de-noise' the figures.

As discussed above, we also decided to remove the streamlines, since this timestep is also visible in Fig. 1. Furthermore, borders are removed as well, leading to a much clearer picture of the phase information, which should be the main focus here. Since the caption does not mention the streamlines, no change is necessary in that regard. The revised figure looks like this:

[Figure]

- L219: "removes small outliers while preserving the shape of a structure" → What is exactly meant by 'small outlier'? Is it thus a size/volume filter, i.e., too small PV features are removed? Some further details on the filtering technique might be helpful.

The morphological operations are filters we do not want to fully introduce in this section, and refer to publications that already do in the manuscript. Still, we agree that the term "outlier" in

this context is not clear from a practical standpoint. Therefore, we rephrase the sentence in lines 218-220 to:

*"A comprehensive overview of these morphological filters is provided by Najman and Talbot (2013), with a recent implementation for extratropical potential vorticity features applied in Fischer et al. (2022). These filters process the 3-D volumetric features by eliminating small, isolated outliers and smoothing out noisy areas, while preserving their general structure."*

- Figure 3: The color-shading, strictly speaking, does not correspond to height (as mentioned in the caption) but to pressure. Provide also in the figure caption the unit of the shading (hPa).

This is correct, the color-shading is by pressure. Therefore, we rephrase the part of the caption of Fig. 3 to:
*In (a), the 0.7 PVU iso-contour is displayed with shading corresponding to atmospheric pressure, identified wave troughs in red, and the trough phase ranging from -pi to 0 is highlighted in blue.*

- Section 2.7: I am not completely sure whether this mini-section is needed, as it is also included in the Data Availability section, or what is missing there you be added.

While we want to emphasize the framework in which the algorithm has been implemented, and also to give it more exposure since it can be used also by other researchers for other identification and tracking tasks, we agree that it does not really justify its own subsection in this context. We remove this small subsection, and add a remark on the framework in the summary section instead.
The summary from line 374 now reads as follows:

*[...] Identified wave troughs and tracks for the entire ERA-5 reanalysis 1940-2022 are provided, along with the framework for identifying and tracking meteorological features (see data availability section). The framework offers support for parallel execution and supports output in JSON or Protobuf formats. Furthermore, a near real-time web-page [...]*

- Figure 6: Here, definitely, the country borders should be removed to make the meteorological fields more prominent; Make also the x-axis labels in the panel 'Occurrence of PV in wavetroughs" consistent to the other panels

The x-axis label should clearly be consistent, this is a mistake on our end. Furthermore, we also as suggested removed the borders in the Figure. No change to the caption is necessary. The revised figure now looks like this:

[Figure]

To keep it consistent, we also removed the country borders from Fig. 7. This is the updated figure:

[Figure]

- L352: I would remove "and previous analyses within the field"; it is rather unspecific.

That is right, we remove that wording from the sentence (Lines 381ff):
"An evaluation of the feature climatology demonstrates the robustness and relevance of our

*approach. The occurrence, extent, and orientation of the identified PV features closely align with established expert knowledge* *."*

- L353: "around identified waves" → "around identified wave troughs"

The paragraph containing this phrase got a rework based on one of the major comments above.

- L376: remove 'valuable'

Line 408: *"and thus to make the*  *PV perspective more easily applicable to AEWs."*

- L356: To what is 'Notable differences' referring back? The sentence before addresses a comparison between PV features and observed/simulated rainfall data. Hence, does 'differences' refer to the comparison between PV and rainfall? This is not completely clear to me.

These "notable differences" refer to the data sources, on one hand the data from forecasts, and on the other hand the data satellite-estimated observations. It specifically refers back to the discussion at lines 278ff. We rephrased the summary to make this reference more clean (Lines 388ff):

*"Notable differences in the spatial distribution and intensity of rainfall* between the simulated and the estimated data can be observed.*"*

**Reviewer # 2:**

Major comments:

One significant concern is that, while the authors do include a review of existing trackers, the introduction lacks references and discussion of more recent trackers. This reviewer suggests that the authors expand on the applications of the trackers already mentioned and incorporate additional references to recent trackers along with their applications.
Below are a few references the authors should include:
AEW tracker by Lawton: https://doi.org/10.1175/MWR-D-21-0321.1
Lawton's AEW tracker applied: https://doi.org/10.1175/MWR-D-23-0005.1
Tropical easterly wave tracks by M. Hollis: https://doi.org/10.1007/s00382-023-07025-w
Alan Brammer's AEW tracker:  https://doi.org/10.1175/MWR-D-15-0106.1
Brammer's AEW tracker applied:  https://doi.org/10.1175/MWR-D-20-0152.1,
https://doi.org/10.1175/JAS-D-20-0339.1
Monsoon low-pressure system tracker by Hurly and Boss 2015:
https://doi.org/10.1002/qj.2447

Another concern is that merely comparing the identified and tracked features to climatology is not sufficient for the proper evaluation and validation of this unique identification and tracking method. This reviewer recommends that the authors incorporate a sensitivity test and compare this identification and tracking scheme to at least one other algorithm or

tracker. A simple sensitivity test, where the authors can compare differences in frequency, intensity, splits, mergers, initiation, and termination, would be highly valuable for the AEW-analyzing community and would support the validity of the proposed technique. The authors do not necessarily need to run the algorithms for the entire climatology, but adding a section that intercompares them for a small period would certainly be insightful. This type of analysis aligns with current trends in the community, where it is becoming more common to evaluate multiple object-based tools. For reference, see Prein et al. 2024 for an MCS intercomparison study: https://doi.org/10.22541/essoar.169841723.36785590/v1. The authors have a unique opportunity here to compare their results to another tracking algorithm.

We appreciate the constructive feedback provided by Reviewer 2.
Firstly, we acknowledge the importance of referencing advancements in tracking methodologies and have therefore expanded our literature review to include additional references such as the AEW trackers developed by Lawton et al. (2022), Hollis et al. (2023) and Brammer and Thorncroft (2015). These inclusions should help to broaden the context for the reader regarding the current state of established wave trackers.

Regarding the comparison of our unique identification and tracking method against existing climatologies and other algorithms, we understand the significance of such analyses. However, the primary aim of this paper was not to compare different 2-D tracking methods but rather to introduce and validate our novel 3-D tracking approach, which builds upon the established 2-D method of Belanger et al. (2016). Our focus is mainly on showcasing the novel aspects and the utility of the 3-D methodology, particularly in how it extends beyond conventional 2-D approaches. Furthermore, while we identify and track *wave trough features*, most other trackers (and AEW identification techniques) in literature operate on *point features*, where a single spatial point represents the position of the wave. This makes a comparative qualitative analysis very challenging. We are not aware of trackers specifically for wave troughs besides the one by Belanger et al., which is based on the idea by Berry et al. (2007), however, they perform manual tracking of the troughs. The trough features allow us to more accurately create composites centered around these troughs.

We agree that a comparative study between 2-D methodologies would be very insightful; however, we believe that comprehensive climatological analyses of the methods are essential for a thorough evaluation. Such a study would significantly exceed the scope of this paper and would surely merit a dedicated publication to do justice to the complexity and depth required for a meaningful comparison. A brief section within this paper would not sufficiently cover the breadth and depth needed for such an analysis.

Nevertheless, we agree with the reviewer that there is a growing need in the literature for objective comparative studies of different methodologies. Therefore, while our paper primarily focuses on the application and results of the established method by Belanger et al., which we have modified and enhanced, we have included references to the suggested comparative works to acknowledge the existence and potential of other methods, as stated above.

While we value the insights and suggestions, here, we want to focus on the questions and topics that align with our research objectives and the scope of our current study. To give the

reader an overview over some existing tracking strategies in literature, we will add the following concise paragraph before introducing our tracking strategy (Lines 140ff):

*"Research on tracking AEWs has produced various methodologies. Hollis et al. (2023), for example, utilize an approach based on the well-known TRACK algorithm, originally proposed by Hodges (1995). This method primarily focuses on linking point features across successive time frames based on a predefined, physically reasonable propagation speed. Lawton et al. (2022) adopt a different approach by tracking AEWs through meridional averages of CV and velocity. Bain et al. (2014) and Brammer and Thorncroft (2015) employ Hovmöller plots for their tracking, analyzing the longitudinal movement of waves.*

*While each of these methodologies has proven effective in their respective applications and has gained popularity in the field, they predominantly focus on point features. In contrast, our work, along with that of Belanger et al. (2015), leverages additional information provided by the identified wave trough features.*

*To form tracks from the identified individual wave troughs, we employ an overlap approach. [...]"*

Newly added references:
Brammer, A., and C. D. Thorncroft, 2015: Variability and Evolution of African Easterly Wave Structures and Their Relationship with Tropical Cyclogenesis over the Eastern Atlantic. Mon. Wea. Rev., 143, 4975–4995, https://doi.org/10.1175/MWR-D-15-0106.1.
Hodges, K. I., 1995: Feature Tracking on the Unit Sphere. Mon. Wea. Rev., 123, 3458–3465, https://doi.org/10.1175/1520-0493(1995)123<3458:FTOTUS>2.0.CO;2.
Hollis, M.A., McCrary, R.R., Stachnik, J.P. et al. A global climatology of tropical easterly waves. Clim Dyn 62, 2317–2332 (2024). https://doi.org/10.1007/s00382-023-07025-w.
Lawton, Q. A., S. J. Majumdar, K. Dotterer, C. Thorncroft, and C. J. Schreck, 2022: The Influence of Convectively Coupled Kelvin Waves on African Easterly Waves in a Wave-Following Framework. Mon. Wea. Rev., 150, 2055–2072, https://doi.org/10.1175/MWR-D-21-0321.1.

Specific comments:

We address the specific comments by Reviewer #2 as follows:

Line 23: AEWs are known to also initiate over high topography, especially over eastern Africa (and thus, the Ethiopian Highlands (Hamilton et al., 2020, Rajarsee et al., 2023 as well as others). Moreover, AEWs that initiate over eastern Africa are more likely to become TC in the eastern Atlantic (i.e., Núñez Ocasio et al., 2021). Please include these references and other relevant references.
&
Line 27: AEWs are also related to the West African offshore rainfall maximum (i.e., Hamilton et al. 2017).

We thank the reviewer for highlighting these additional valuable resources and for drawing attention to the initiation areas of AEWs that were previously omitted. Acknowledging the Ethiopian Highlands as a significant origin point for these waves is indeed useful in this context. While our focus does not lie in the specific likelihood of AEWs originating from eastern Africa developing into tropical cyclones, citing these references underscores the

depth and breadth of research surrounding AEW dynamics. In response to your suggestions, we have updated the manuscript to include these areas and references (Lines 20ff):

*"African Easterly Waves (AEWs) are synoptic-scale disturbances that play a crucial role in the weather and climate of tropical West Africa and the tropical Atlantic region. AEWs are quasi-periodic perturbations, ==typically originating over the broader Lake Chad region in central North Africa or being triggered by the high topography over the Ethiopian Highlands (Mekonnen et al., 2006, Hamilton et al., 2020). These waves propagate== westward across West Africa, the North Atlantic Ocean, and as far as the eastern Pacific.*
*They have drawn considerable attention due to their substantial impact on Atlantic tropical cyclone (TC) genesis ==(e.g., Russell et al., 2017; Núñez Ocasio et al., 2021; Rajasree et al., 2023), rainfall variability over the West African monsoon region, their relation to the West African offshore rainfall maximum (Hamilton et al., 2017), and their role in extreme precipitation events over tropical West Africa== (e.g., Fink and Reiner, 2003; Crétat et al., 2015; Engel et al., 2017). Understanding the formation, propagation, and interaction of AEWs with the ambient (thermo-)dynamical state of the troposphere is essential for improving the prediction of North Atlantic TCs and rainfall patterns in West Africa."*

Newly added references:
**Hamilton**, H. L., G. S. Young, J. L. Evans, J. D. Fuentes, and K. M. Núñez Ocasio (2017), The relationship between the Guinea Highlands and the West African offshore rainfall maximum, Geophys. Res. Lett., 44, 1158–1166, doi:10.1002/2016GL071170.

**Hamilton**, H. L., Núñez Ocasio, K. M., Evans, J. L., Young, G. S., & Fuentes, J. D. (2020). Topographic influence on the African Easterly Jet and African Easterly Wave energetics. Journal of Geophysical Research: Atmospheres, 125, e2019JD032138. https://doi.org/10.1029/2019JD032138
**Mekonnen**, A., C. D. Thorncroft, and A. R. Aiyyer, 2006: Analysis of Convection and Its Association with African Easterly Waves. J. Climate, 19, 5405–5421, https://doi.org/10.1175/JCLI3920.1.
**Núñez Ocasio**, K. (2021). Tropical cyclogenesis and its relation to interactions between African easterly waves and mesoscale convective systems.
**Rajasree**, V. P. M., et al. "Tropical cyclogenesis: Controlling factors and physical mechanisms." Tropical Cyclone Research and Review (2023).

Line 53: Please rephrase "theoretical perspective" as the other references also incorporate theories in their analysis. Dunkerton et al., 2009 proposed a simpler 'geometrical', forecaster-friendly idea.

We agree, in this context the "theoretical perspective" should be omitted since it is not unique to the cited reference there; especially after adding the references above.
Line 59: *" Dunkerton et al. (2009) provide a conceptual framework that links AEWs and TC genesis. [...]"*

Tracking: It is unique that the authors incorporate MCS tracking methods into the tracking of AEWs. Both overlapping (Houze, Evans, and Shemo, PyFLEXTRKR by Z. Feng, TAMS by Núñez Ocasio, and MOAAP by Andreas Prein) and graph theory (Kim Whitehall) are used to track MCSs. Please make a point of this and include references.

We thank the reviewer for pointing us to these relevant references for existing tracking techniques. Although the tracking of 2-D wave troughs is not the central focus of our work, we acknowledge its significance and have opted for a more concise presentation without

extensive comparisons to existing methods. Nonetheless, we agree that it is essential to acknowledge the contributions from both MCS and AEW tracking methods, as we use ideas from both worlds. Following Reviewer #1's suggestion, we have revised the introduction to the tracking section for clearer guidance on how wave troughs are tracked through overlap, ensuring a straightforward path to understanding the methodology employed. The revised tracking section now begins as follows (Lines 149ff):

*"To form tracks from the identified individual wave troughs, we employ an overlap approach. Overlap tracking has proven to be a robust tracking technique in meteorological applications, such as tracking of Mesoscale Convective Systems (Núñez Ocasio et al, 2020; Feng et al. 2023; Prein et al., 2023) and general purpose feature extraction (Ullrich et al., 2021). However, since our identified wave troughs are represented as line strings, they don't directly lend themselves to traditional overlap tracking methods.*

*To address this, we create area features by predicting the future positions of each trough for upcoming time steps, t+Δt and t+2Δt, with Δt set to 6 hours. This prediction uses an anticipated propagation speed to define a polygonal area that represents where the trough is expected to be. The presence of overlap between these predicted polygonal areas and the actual locations of wave troughs at future time steps facilitates the tracking. The range of expected propagation speeds, which we set from $u\_max=-15ms^{-1}$ to $u\_min=+2ms^{-1}$, defines the size and shape of these polygons, as illustrated in Fig. 1b."*

*[...]*

*Using the identified connections, a connection graph $G=(W,E)$ is formed, akin to the approach outlined by Limbach et al. (2012). This structure enables the application of graph theory concepts and algorithms to our identified features, as also employed by Whitehall et al. (2015) in their analysis of MCS. W represents the set [...]"*

Newly added references:
**Feng**, Z., Hardin, J., Barnes, H. C., Li, J., Leung, L. R., Varble, A., & Zhang, Z. (2023). PyFLEXTRKR: a flexible feature tracking Python software for convective cloud analysis. Geoscientific Model Development, 16(10), 2753-2776.
**Prein**, A. F., Mooney, P. A., & Done, J. M. (2023). The multi-scale interactions of atmospheric phenomenon in mean and extreme precipitation. Earth's Future, 11, e2023EF003534. https://doi.org/10.1029/2023EF003534
**Ullrich**, P. A., Zarzycki, C. M., McClenny, E. E., Pinheiro, M. C., Stansfield, A. M., and Reed, K. A.: TempestExtremes v2.1: a community framework for feature detection, tracking, and analysis in large datasets, Geosci. Model Dev., 14, 5023–5048, https://doi.org/10.5194/gmd-14-5023-2021, 2021.
**Whitehall**, K., Mattmann, C.A., Jenkins, G. et al. Exploring a graph theory based algorithm for automated identification and characterization of large mesoscale convective systems in satellite datasets. Earth Sci Inform 8, 663–675 (2015). https://doi.org/10.1007/s12145-014-0181-3

Line 150: What features do the authors refer to? It is more likely for convection or PV features to split and merge than the actual AEW splitting and merging. Please clarify.

In this section, we are focusing on identifying and tracking the AEW troughs, without taking into account any PV view or analysis. The splitting and merging mentioned here are indeed the AEW troughs. While it is unlikely for features to split or merge in the sense of one feature "catching up" to another, we sometimes observe a split into a northern and southern track. The northern track might recurve into the Atlantic, while the southern track stays intact and

propagates westward. Features also sometimes split temporarily due to a weakening in the vorticity signal in the middle (compared to a stronger signal at the northern and southern end of the wave trough, as observable in Fig. 1c at some points in time. We clarified this by rephrasing lines 179-181 as follows:

*"The extraction of tracks poses challenges: During a wave trough's life cycle, it interacts with the dynamic environment and might split into multiple parts (e.g., due to a weakening at the center part of the trough), and potentially merge again later."*

Line 206: And how do you account for these artifacts

In hindsight, we think the term "artifacts" does not clearly state what we were intended to state here. In regions not favorable for AEW activity, signals in the chosen frequency band of 2-8 days mostly do not have an origin in the AEW signal. Other signals (other tropical waves, MCS signals…) become more dominant in these regions, leading to higher noise and less robust results in the phase field. Therefore, we rephrase that part in line 238ff to:

*"In areas less conducive to AEW activity, the Hilbert transform's reliability diminishes, leading to increased noise and less accurate results in the phase field. This outcome is anticipated, as different types of atmospheric waves and other atmospheric disturbances overshadow the AEW signal in these locations."*

Figure 3: Is shading height or PVU contours? Please clarify and add units.

The definition of the shading, as also pointed out by Reviewer #1, is unclear. It is actually shaded by pressure levels to give a better perception of height. Therefore, we revised the caption to:
*"In (a), the 0.7 PVU iso-contour is displayed with shading corresponding to atmospheric pressure, identified wave troughs in red, and the trough phase ranging from -pi to 0 is highlighted in blue."*

Line 227: As well as thermodynamic constraints (Núñez Ocasio and Rios Berrios 2023). Please add.

As proposed we added the information that the structure of PV features also depend on thermodynamic constraints (Lines 261ff):
 *"Previous studies have revealed that the structure and orientation of PV features in AEWs depend on different factors, for example their location and the (thermo-)dynamic constraints of the environment (Tomassini et al., 2017; Russell et al., 2020; Núñez Ocasio and Rios-Berrios 2023)."*

Newly added references:
**Núñez Ocasio**, K. M., & Rios-Berrios, R. (2023). African easterly wave evolution and tropical cyclogenesis in a pre-Helene (2006) hindcast using the Model for Prediction Across Scales-Atmosphere (MPAS-A). Journal of Advances in Modeling Earth Systems, 15, e2022MS003181. https://doi.org/10.1029/2022MS003181

Lines 360: The AEJ was not studied in detail and so it is a speculation. Remove or add such analysis.

As illustrated at Page 6 of this review document, we added the meridional wind component to the longitude, and the zonal wind component to the latitude cross-section composites based on a suggestion from Reviewer 1. This also gives us the opportunity to investigate the AEJ relative to the wave troughs. The latitude cross section shows a maximum of zonal wind just north of the wave centers. Especially over the ocean, this maximum of the jet clearly collocates with the blue PV anomaly:

[Figure]

We also changed the passage interpreting the figure above (Lines 339ff):

*"[...] Over land, the PV column is upright, extending higher in the vertical, signifying intense deep convection. Over the ocean, anomalous PV to the north of the features can be observed,* especially in the 600-700 hPa range, which coincides with the peak intensity zone of the AEJ. The contours in the Figure distinctly mark the core of the AEJ just north of the wave centers. *This lets us suggest that the PV anomaly in the composite can be traced back to PV advection taking place from tropical West Africa to the Atlantic Ocean."*

We suggest that the addition of the u-wind to the plot and the added description should be sufficient to trace back the PV anomaly to the AEJ, and therefore to keep the statement in question in the conclusion.

---

## Author Response (AR2)

**RESPONSE TO REVIEWERS**
**An objective identification technique for potential vorticity structures associated with African Easterly Waves**
**gmd-2023-218**

We thank the reviewers and the topical editor for accepting the submitted manuscript. The reviews have been instrumental in enhancing the overall quality of our submission and allowing us to refine the presented arguments.

Following the feedback provided by Reviewer #2, we have updated our bibliography to include the recommended references.